# VVC-Gym: A Fixed-Wing UAV Reinforcement Learning Environment for Multi-Goal Long-Horizon Problems

**Xudong Gong**[1,2]   **Dawei Feng**[1,2*]   **Kele Xu**[1,2]   **Weijia Wang**[3]   **Zhangjun Sun**[3]
**Xing Zhou**[4]   **Si Zheng**[5]   **Bo Ding**[1,2]   **Huaimin Wang**[1,2]

[1] College of Computer Science and Technology, National University of Defense Technology, Changsha, Hunan, China
[2] State Key Laboratory of Complex & Critical Software Environment, Changsha, Hunan, China
[3] Flight Automatic Control Research Institute, AVIC, Xian, Shaanxi, China
[4] College of Intelligence Science and Technology, National University of Defense Technology, Changsha, Hunan, China
[5] Qiyuan Lab, Beijing, China

## Abstract

Multi-goal long-horizon problems are prevalent in real-world applications. The additional goal space introduced by multi-goal problems intensifies the spatial complexity of exploration; meanwhile, the long interaction sequences in long-horizon problems exacerbate the temporal complexity of exploration. Addressing the great exploration challenge posed by multi-goal long-horizon problems depends not only on the design of algorithms but also on the design of environments and the availability of demonstrations to assist in training. To facilitate the above research, we propose a multi-goal long-horizon Reinforcement Learning (RL) environment based on realistic fixed-wing UAV's velocity vector control, named VVC-Gym, and generate multiple demonstration sets of various quality. Through experimentation, we analyze the impact of different environment designs on training, assess the quantity and quality of demonstrations and their influence on training, and assess the effectiveness of various RL algorithms, providing baselines on VVC-Gym and its corresponding demonstrations. The results suggest that VVC-Gym is suitable for studying: (1) the influence of environment designs on addressing multi-goal long-horizon problems with RL. (2) the assistance that demonstrations can provide in overcoming the exploration challenges of multi-goal long-horizon problems. (3) the RL algorithm designs with the least possible impact from environment designs on the efficiency and effectiveness of training. Our code is available at https://github.com/GongXudong/fly-craft and https://github.com/GongXudong/fly-craft-examples and demonstrations are available at https://www.openml.org/d/46000.

## 1 Introduction

Many real-world applications fall into the category of multi-goal long-horizon problems. For instance, a UAV must be capable of achieving not only the left-side goal but also the right-side goal (multi-goal); when completing an ascending turn, it is necessary to perform a horizontal turn first, then accelerate in a straight line, and finally climb in altitude (long-horizon). Addressing multi-goal long-horizon problems with Reinforcement Learning (RL) (Sutton & Barto, 2018) encounters a significant exploration challenge: (1) the multi-goal nature requires the policy to explore not only the state and action spaces but also the additional goal space during training, which intensifies the spatial complexity of exploration; (2) due to the learning signal decreasing exponentially with the horizon (Osband et al., 2016), the long-horizon nature exacerbates the temporal complexity of exploration.

---

*Correspondence to Dawei Feng `davyfeng.c@qq.com`.

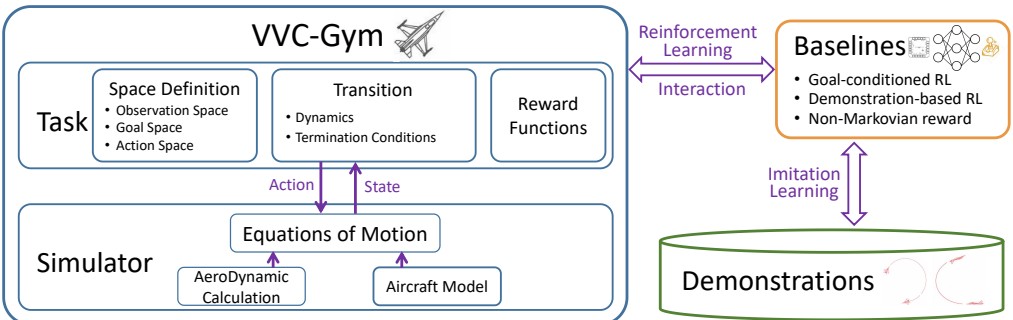

Figure 1: The framework of VVC-Gym.

Existing work predominantly focuses on the design of algorithms, neglecting the importance of environment design and the potential benefits that demonstrations can provide during training. Specifically, (1) insufficient environment designs hinder training efficiency and effectiveness, such as the absence of effective termination conditions, which can lead to the collection of numerous ineffective samples after severe states occur; (2) demonstrations can assist in overcoming the exploration challenge from multiple perspectives, for instance, by constraining the policy's exploration space and providing a behavioral prior for the policy. Moreover, within the Goal-Conditioned RL (GCRL) community (Liu et al., 2022), existing multi-goal environments (Plappert et al., 2018; Ghosh et al., 2020; Liu et al., 2022; Hu et al., 2022) often feature tasks that are relatively simple with short control sequences, while also commonly encountering the aforementioned two issues. Consequently, the GCRL community urgently requires a multi-goal long-horizon environment that allows researchers to (1) evaluate GCRL on more realistic and complex tasks, (2) investigate the impact of environment designs on GCRL training, (3) investigate how demonstrations can be utilized to overcome the exploration challenge, and (4) when engaged in algorithm design, suffer the least possible impact from environment design on the efficiency and effectiveness of training.

To facilitate the community's research on multi-goal long-horizon problems, we: (1) provide the GCRL community with the first RL environment on realistic fixed-wing UAV's velocity vector control (VVC) task, VVC-Gym. VVC is a typical multi-goal long-horizon problem; on one hand, VVC requires the UAV to achieve arbitrary desired velocity vectors, which is a typical multi-goal problem. On the other hand, fixed-wing UAVs are high-speed vehicles, and precise control at high velocities necessitates frequent control outputs (Wang & Wang, 2020), resulting in the long-horizon problem. (2) conduct ablation studies on the environment design of VVC-Gym, showcasing how VVC-Gym can facilitate researchers in studying the influence of environment design on GCRL training. (3) Equip VVC-Gym with multi-quality demonstration sets, evaluate them from various perspectives, and show their effectiveness in aiding GCRL in overcoming the exploration challenge. (4) Provide baselines on VVC-Gym and corresponding demonstrations to illustrate their broad applicability in studying RL, including GCRL, demonstration-based RL (Ramírez et al., 2022).

## 2 RELATED WORK

**Multi-goal environments.** Existing GCRL research often conducts experiments on open-source multi-goal environments, such as: (1) the robotic arm control task based on Mujoco (Nair et al., 2018; Plappert et al., 2018; Gupta et al., 2020; Foundation, 2022), DMC (Tunyasuvunakool et al., 2020), Panda-Gym (Gallouédec et al., 2021). (2) maze-like tasks such as PointMaze (Trott et al., 2019) and AntMaze (Nachum et al., 2018). (3) Atari games (Warde-Farley et al., 2018), etc. Most of these environments either utilize non-realistic tasks with short horizons or employ simplistic environment designs, such as sparse rewards, without considering mechanisms for terminating episodes in the event of severe states, resulting in low exploration efficiency at the environment level. Additionally, some lack accompanying demonstrations, which precludes the support for demonstration-based RL. In contrast, we introduce VVC-Gym to address these issues. VVC-Gym features various environment designs and includes accompanying demonstration sets. It facilitates studying the influence of environment designs on GCRL training and demonstrations' aids in overcoming exploration

challenges. Additionally, the default environment designs of VVC-Gym enable effective GCRL algorithm evaluation unaffected by environmental designs.

**RL for multi-goal problems.** Hindsight experience replay (HER) (Andrychowicz et al., 2017) is a core method employed by existing GCRL algorithms (GCRL algorithm background is provided in Appendix E) to address the multi-goal challenge. HER enhances sample efficiency by replacing the desired goals of failed trajectories with the achieved states to yield positive rewards. (Pitis et al., 2020) summarizes a general goal-conditioned off-policy RL framework for solving multi-goal problems. In this framework, most research focuses on how to sample behavioral goals to further improve sampling efficiency, with the core idea being to avoid sampling training data on goals that the current policy has no capability of achieving (also known as self-curriculum methods). For instance, RIG (Nair et al., 2018) samples goals directly from the distribution of achieved goals, DISCERN (Warde-Farley et al., 2018) samples uniformly on the support set of the distribution of achieved goals, and MEGA (Pitis et al., 2020) uses inverse probability weighting sampling (Jacq et al., 2023) on the distribution of achieved goals to samples goals that the current policy can achieve but not well. Additionally, (Gong et al., 2024a) introduces a general framework for goal-conditioned on-policy RL algorithms. These studies focus on the algorithmic perspective, without considering the potential impact of environment design on training efficiency.

**RL for long-horizon problems.** Existing work addresses the long-horizon challenges through three approaches. The first approach is Imitation Learning (IL) (Zheng et al., 2022), where policies learned from demonstrations serve as a behavioral prior to bias the exploration of RL (Baker et al., 2022; Gong et al., 2024b). This method implicitly constrains the exploration space, thereby significantly enhancing exploration efficiency (Ramírez et al., 2022). The second approach involves designing efficient exploration strategies, with a focus on the behavioral goal selection strategy (refer to the above RL for multi-goal problems). The third approach is to design subgoal generation strategies, which are centered around the idea of decomposing long-horizon tasks into smaller, more manageable subtasks (Nasiriany et al., 2019). While the aforementioned research concentrates on algorithms designs to tackle the long-horizon challenge, our work is dedicated to providing the GCRL community with a novel and realistic multi-goal long-horizon RL environment. We offer baselines for the three aforementioned methods on VVC-Gym, demonstrating its suitability as a testbed for researching multi-goal long-horizon problems.

## 3 VVC-GYM

The framework of VVC-Gym is depicted in Fig. 1. VVC-Gym comprises two main modules: the Simulator and the Task. The Simulator is responsible for UAV-related simulations, while the Task encapsulates the velocity vector control task based on the Gymnasium. Additionally, we equip VVC-Gym with multi-quality demonstrations and provide baselines on VVC-Gym. The detailed simulation process is described in Appendix A, and the baselines are presented in Section 4.3. In this section, we provide a detailed introduction to the Task and demonstrations.

### 3.1 PROBLEM FORMULATION

During the process, roll angle $\phi$, pitch angle $\theta$, yaw angle $\psi$, flight path elevator angle $\mu$, flight path azimuth angle $\chi$, true airspeed $v$, altitude $h$, roll rate $p$ (observation space) are observable. The agent is tasked with manipulating the UAV's velocity vector $(v, \mu, \chi)$ to match a desired velocity vector $(v_g, \mu_g, \chi_g)$ (goal space), by controlling the deflections of the aileron actuator $\delta_a$, elevator actuator $\delta_e$, rudder actuator $\delta_r$, and power level actuator $\delta_{pla}$ (action space).

Two control modes are provided: The first is the guidance law mode, where the controller outputs intermediate control commands: roll rate command $p_c$, overload command $n_{zc}$, and the position of the power level actuator $\delta_{pla}$. These three intermediate commands are then transformed into final control commands $\delta_a, \delta_e, \delta_r, \delta_{pla}$ through a control law model that smooths the output, which is detailed in Appendix B. The smoothing action of the control law model serves a purpose similar to the temporal abstraction provided by frame-skipping (Kalyanakrishnan et al., 2021), alleviating the temporal complexity of exploration. The second mode is the end-to-end mode, where the controller directly outputs $\delta_a, \delta_e, \delta_r, \delta_{pla}$. This mode exhibits significant oscillation in actions during training, and combined with the long-horizon nature of the VVC task, the training suffers greatly from the

temporal complexity of exploration. In Appendix I, we provide experimental analysis of the distinct impacts of these two training modes on RL training.

## 3.2 TRANSITION

The VVC-Gym's Transition employs a modular design principle, comprising the Dynamics sub-module and the Termination Condition sub-module. The Dynamics sub-module is responsible for communicating with the Simulator, sending actions, and receiving the new state of the UAV. The Simulator performs the core computations of the aerodynamic equations detailed in Appendix A. The Termination Condition sub-module is tasked with terminating an episode based on certain conditions, which are user-defined and can be used to study how to avoid collecting ineffective samples, thereby enhancing exploration efficiency. To facilitate research, we provide 3 categories, totaling 7 default termination conditions:

The first category consists of termination conditions for determining whether the UAV has achieved its desired goals:

(1) **R**each **T**arget Termination (RT): if error of $\vec{v}$ keeps in the error band for at least $T_R$ steps.

The second category consists of termination conditions for truncating excessively long trajectories from both temporal and spatial dimensions:

(2) **T**imeout termination (T): if the goal is not achieved within the step limit $T_{max}$.
(3) **C**rash Termination (C): if the altitude of UAV is less than the safe altitude $h_0$.

The third category consists of termination conditions for truncating unreasonable or invalid trajectories:

(4) **C**ontinuously **M**ove **A**way Termination (CMA): if the velocity vector moves away from the desired velocity vector continuously for $T_M$ steps.
(5) **C**ontinuously **R**oll Termination (CR): if the UAV rolls $\phi_{max}$ continuously.
(6) **E**xtreme **S**tate Termination (ES): if the true airspeed $v$ reaches the limit $v_{max}$ or the roll rate $p$ reaches the limit $p_{max}$.
(7) **N**egative **O**verload and **B**ig **R**oll Angle Termination (NOBR): if the overload component along the OZ axis of the Body Coordinate System $n_z$ is negative and the roll angle $\phi$ is bigger than a limit $\phi_N$ for more than $T_N$ steps.

We believe that for multi-goal long-horizon problems, termination conditions can be designed by referencing the above three categories, which respectively represent (1) the conditions for goal achievement, (2) the maximum limit for exploration, and (3) the truncation of unreasonable or invalid trajectories based on the characteristics of the task itself. The 7 termination conditions proposed for our fixed-wing UAV control scenario are all relatively simple. Although they are straightforward, our experiments in Section 4 show that they can significantly enhance exploration efficiency.

## 3.3 REWARD FUNCTION

Addressing multi-goal long-horizon problems' exploration challenge requires informative rewards, which can remarkably enhance training efficiency(Sutton & Barto, 2018; Eckstein & Schiffmann, 2020). Existing multi-goal long-horizon environments typically employ either sparse rewards or simplistic distance-based rewards, such as $r_g(s_t) = -\|\zeta(s_t), g\|$, where $\zeta$ maps a state to its achieved goal and $\|\cdot\|$ calculates the difference between two goals (Liu et al., 2022). Such simplistic reward functions are not conducive to investigation into the impact of reward design on training. To this end, we propose a general distance-based goal-conditioned reward structure:

$$r_{g,t} = \begin{cases} 0, & \text{if triggers RT} \\ r_{penalty}, & \text{if triggers any of CMA, CR, C, ES, or NOBR} \\ -(\frac{\|\zeta(s_t)-g\|}{\sigma})^b, & \text{else} \end{cases} \quad (1)$$

where $r_{penalty}$ is a negative number, $\|.\|$ calculates the difference in two velocity vectors, $\sigma$ is a normalization factor for velocity vector difference such that $\frac{\|\zeta(s_t)-g\|}{\sigma} \in [0, 1]$, $b$ is a scaling factor

that controls change rate of the reward function over different error intervals. This reward function gives the agent a negative reward between $[-1, 0]$ at each step if the agent does not trigger any termination condition. The closer the current velocity vector aligns with the desired velocity vector, the closer the reward gets to 0. This negative reward spurs the agent to achieve the desired velocity vector promptly. The differences from previous rewards (Bøhn et al., 2019; Koch et al., 2019b;a) are highlighted in two aspects:

**(1) Penalty $r_{penalty}$** is incorporated into the reward to work in synergy with the termination conditions, thereby enhancing exploration efficiency. It penalizes the agent for activating non-RT terminations. Crucially, $r_{penalty}$ must be much less than the minimum single-step reward to prevent premature episode terminations. This ensures the agent doesn't intentionally trigger termination to avoid low cumulative reward due to negative rewards in each step (Eckstein & Schiffmann, 2020).

We propose two approaches to set $r_{penalty}$: The **large negative constant** method, which sets $r_{penalty}$ much lower than the average reward from triggering termination T, to discourage premature episode terminations via other conditions. The **pessimistic estimation** method, which sets $r_{penalty}$ based on the remaining steps until the maximum $T_{max}$. If the agent triggers a termination at step $T_\tau$ and receives the worst reward of $-1$ per step, $r_{penalty}$ is given by $r_{penalty} = -\frac{1-\gamma^{T_{max}-T_\tau}}{1-\gamma}$. We analyze the effects of these methods on learning in Section 4.4.2.

**(2) Scaling factor $b$** is incorporated into the reward to adjust the change rate of rewards in different learning stages. As shown in Fig. 2, with $b > 1$, the reward reacts quickly to large errors but slowly to small ones, speeding up goal approach from afar but hindering fine adjustments near the goal. Conversely, $b < 1$ favors accuracy over speed. The choice of $b$ balances the speed of goal achievement with its accuracy. The effect of $b$ on learning is analyzed in Section 4.4.2.

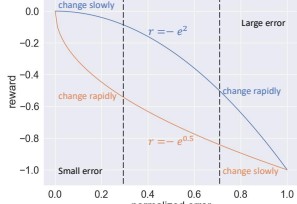

Figure 2: reward b

### 3.4 DEMONSTRATIONS

Demonstrations facilitate the utilization of demonstration-based methods to enhance the learning efficiency of RL, enabling RL to tackle exploration-challenging tasks (Ramírez et al., 2022). When collecting demonstrations, it is significantly less labor-intensive for human experts to design a simple controller than to collect directly. However, this approach presents the challenge of low demonstration quality (Brown et al., 2020; Sasaki & Yamashina, 2020; Xu et al., 2022) from the simple controller designs. Nevertheless, demonstrations generated by human-designed simple controllers still hold significant value in enhancing the efficiency of RL training (Gong et al., 2024b). Therefore, based on a simple human-designed PID controller (Visioli, 2006), detailed in Appendix C, we generate multi-quality demonstrations through the following three steps:

**Generating demonstrations with the PID controller**. We first discretize the desired goal space, detailed in Appendix D, to obtain a desired goal set. We then use the PID controller to sample within this goal set, recording trajectories that successfully trigger the RT condition as $\mathcal{D}_E^0$.

**Augmenting demonstrations based on symmetry**. Demonstration $\mathcal{D}_E$ is augmented by leveraging the symmetry in the flight path azimuth angle, as detailed in Appendix D. The augmented demonstration set is denoted as $\overline{\mathcal{D}_E}$.

**Generating more and high-quality demonstrations**. Iterative Regularized Policy Optimization (IRPO) (Gong et al., 2024b) is employed to iteratively optimize the policy and generate demonstrations $\mathcal{D}_E^i, i \in [1, 2, \ldots, N_\mathcal{D}]$, with increasingly improved quantity and quality. These demonstrations are then further augmented with the method detailed in the second step, resulting in $\overline{\mathcal{D}_E^i}, i \in [1, 2, \ldots, N_\mathcal{D}]$.

In Section 4.2, we provide a detailed analysis of the differences between these demonstration sets.

## 4 EXPERIMENTS

In this section, we: (1) compare the impact of using versus not using the environment designs introduced in Section 3 on GCRL training. (2) present analysis on demonstration quantity and quality. (3) provide baselines for GCRL and demonstration-based RL, showcasing the extensive

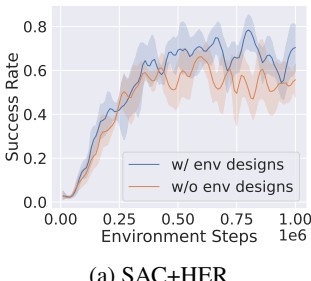 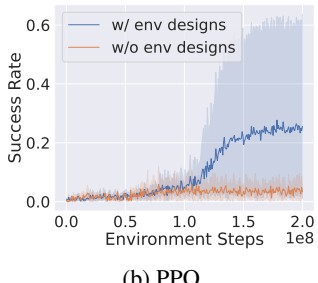 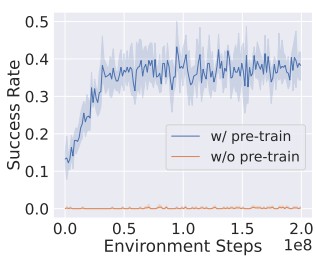

(a) SAC+HER  (b) PPO

Figure 3: Comparison between policies trained by SAC+HER and PPO on using versus not using the environment designs introduced in Section 3. "w/ env designs" refers to using all 7 termination conditions and reward Eq. 1, while "w/o env designs" refers to only using RT and T and reward $r_g(s_t) = -\|\zeta(s_t), g\|$. Results come from experiments over 5 random seeds.

Figure 4: GCRL with and without pre-training from demonstrations. Results come from PPO and $\mathcal{D}_E^0$ trained on the hard version of VVC-Gym over 5 random seeds.

applicability of VVC-Gym and demonstrations in GCRL research. (4) conduct ablation studies to assess the impact of various environment designs on GCRL training.

In the experiments, we employ two versions of the velocity vector control task: the normal version and the hard version, which are detailed in Appendix F. In the experiments, unless otherwise specified, the normal version are used, with the hard version only used when specifically indicated.

## 4.1 MAIN RESULTS

To demonstrate the benefits of the termination conditions and informative reward function in addressing multi-goal long-horizon problems, we evaluate the impact of the environment designs proposed in Section 3 on GCRL training by training policies with SAC (Haarnoja et al., 2018) + HER (Andrychowicz et al., 2017) and PPO (Schulman et al., 2017) (with a goal-conditioned policy (Schaul et al., 2015)) on VVC-Gym. The experimental results are displayed in Fig. 3. The results indicate that, irrespective of the specific algorithm employed, policies trained with our environment designs surpass those trained without these designs, when an equivalent training budget is allocated. This demonstrates that the termination conditions and reward function tailored for VVC-Gym effectively facilitate more efficient training for GCRL algorithms. We provide a detailed analysis of the reasons why the termination conditions and reward function contribute to the training in Section 4.4.

To illustrate the benefits of demonstrations for GCRL training, we examine the effect of pre-training with demonstrations on the subsequent GCRL training. The results are presented in Fig. 4. It is evident that when an algorithm attempts to train a policy directly without pre-training, it struggles to learn any skills due to the task's intrinsic exploration complexity. In contrast, after pre-training the policy with Goal-Conditioned Behavioral Cloning (GCBC) (Ding et al., 2019), the algorithm is capable of further improving policy performance through subsequent GCRL training. This highlights the importance of demonstrations in facilitating more efficient GCRL training. An extensive analysis of demonstration quantity and quality is provided in Section 4.2, and the performance of various algorithms on different demonstration sets is detailed in Section 4.3.

## 4.2 ANALYSIS OF DEMONSTRATION QUANTITY AND QUALITY

Demonstrations are collected with methods detailed in Section 3.4. IRPO is used to collect $N_{\mathcal{D}} = 3$ demonstration sets. We analyze the statistical information of the demonstration sets, and the results are presented in Table 1. In terms of quantity, as the index $i$ increases, $\mathcal{D}_E^i$ covers an increasing number of goals, resulting in an increase in total transitions. In terms of quality, there is no significant difference in the goal achieving accuracy. In addition, as discussed in Appendix M, we also consider the state and action smoothness, which also do not exhibit significant differences across different $\mathcal{D}_E^i$. However, the average length of the trajectories decreases, indicating that as $i$ increases, the trajectories contained within $\mathcal{D}_E^i$ are able to achieve goals more quickly without compromising goal achieving accuracy and state and action smoothness. In summary, as $i$ increases, the trajectory quantity and quality of $\mathcal{D}_E^i$ improves. The variations among these 8 demonstration sets enable re-

Table 1: Trajectory Number, Average Length of Trajectory, Total Transition Number, and Goal Achieving Accuracy of Demonstrations

| Demonstration | Number of trajectories | Goal space coverage (%) | Average length of trajectories | Number of transitions | Accuracy | | |
|---|---|---|---|---|---|---|---|
| | | | | | $error_v$ | $error_\mu$ | $error_\chi$ |
| $\mathcal{D}_E^0$ | 10184 | 20.08 | 282.01±149.98 | 2872051 | 6.56±3.25 | 0.36±0.35 | 0.53±0.45 |
| $\overline{\mathcal{D}_E^0}$ | 10264 | 20.24 | 281.83±149.48 | 2892731 | 6.56±3.25 | 0.36±0.36 | 0.53±0.45 |
| $\mathcal{D}_E^1$ | 24924 | 49.15 | 124.64±53.07 | 3106516 | 4.12±3.45 | 0.59±0.32 | 0.57±0.41 |
| $\overline{\mathcal{D}_E^1}$ | 27021 | 53.28 | 119.64±47.55 | 3232896 | 4.47±3.49 | 0.58±0.32 | 0.60±0.44 |
| $\mathcal{D}_E^2$ | 33114 | 65.29 | 117.65±46.24 | 3895791 | 4.83±3.45 | 0.57±0.33 | 0.66±0.54 |
| $\overline{\mathcal{D}_E^2}$ | 34952 | 68.92 | 115.76±45.65 | 4045887 | 5.16±3.47 | 0.56±0.33 | 0.68±0.60 |
| $\mathcal{D}_3$ | 38654 | 76.22 | 116.59±46.81 | 4506827 | 5.24±3.41 | 0.60±0.34 | 0.71±0.69 |
| $\overline{\mathcal{D}_E^3}$ | 39835 | 78.55 | 116.56±47.62 | 4643048 | 5.29±3.38 | 0.60±0.35 | 0.74±0.75 |

Table 2: Performance of GCRL algorithms on VVC-Gym. Results (% success rates over 5 random seeds) come from the hard version of VVC-Gym.

Table 3: Performance of pre-train & fine-tune algorithms on the hard version of VVC-Gym (% success rates over 5 random seeds).

(a) GCRL algorithms

| RL type | Algorithm | Success rate |
|---|---|---|
| Off-policy | SAC | 1.08±0.48 |
| | HER | 8.32±1.86 |
| On-policy | PPO | 0.04±0.03 |
| | GCBC + PPO | 38.31±1.62 |

(b) Curriculum methods

| Curriculum | Success rate |
|---|---|
| None | 38.31±1.62 |
| RIG | 49.03±1.54 |
| DISCERN | 49.36±1.91 |
| MEGA | 48.62±2.35 |

| Demos | GCBC | GCBC + PPO |
|---|---|---|
| $\overline{\mathcal{D}_E^0}$ | 17.08±0.57 | 38.31±1.62 |
| $\overline{\mathcal{D}_E^1}$ | 36.54±1.97 | 53.83±0.80 |
| $\overline{\mathcal{D}_E^2}$ | 41.79±0.44 | 68.47±1.20 |
| $\overline{\mathcal{D}_E^3}$ | 42.77±1.35 | 71.68±2.86 |

searchers to investigate the influence of demonstration quantity and quality on demonstration-based RL.

### 4.3 GCRL AND DEMONSTRATION-BASED RL BASELINES

In this section, we show the potential of VVC-Gym in studying RL by establishing baselines for: (1) GCRL. (2) demonstration-based RL, including IL and RL from demonstrations (RLfD) (Ramírez et al., 2022). Additionally, the reward function in VVC-Gym can be configured as Non-Markovian Reward (NMR) (Abel et al., 2021). We discuss NMR in Appendix H and provide relevant baselines.

**Investigating GCRL.** To demonstrate the suitability of VVC-Gym for studying GCRL, we conduct experiments on VVC-Gym as follows:

(1) We train the off-policy GCRL algorithm SAC and HER, the on-policy GCRL algorithm PPO and GCBC+PPO, with the corresponding results presented in Table 2a. It is noted that, firstly, while employing a goal-conditioned policy or a goal-conditioned value function directly can adapt standard RL algorithms (SAC and PPO) to address multi-goal problems, the performance is poor. However, the integration of techniques specifically designed to tackle exploration challenges, such as HER and GCBC-pretraining, leads to a significant improvement in policy performance. Secondly, both HER and GCBC+PPO show potential for increased success rates, suggesting that VVC-Gym poses a challenging task that is well-suited for studying multi-goal long-horizon problems.

(2) We train self-curriculum methods including RIG (Nair et al., 2018), DISCERN (Warde-Farley et al., 2018), and MEGA (Pitis et al., 2020), with the results presented in Table 2b. It is evident that different self-curriculum methods can enhance learning effectiveness, although the differences among these three methods are not pronounced, suggesting that VVC-Gym is suitable for studying self-curriculum in GCRL. It has been observed that while the implementation of curriculum methods can significantly enhance the policy's success rate, the rate remains relatively low for the VVC task. This is primarily attributed to the inherent challenges associated with the VVC task, which we have thoroughly discussed in Appendix J.

(3) We utilize the policy trained by GCBC+PPO+MEGA (Gong et al., 2025) to complete a long-horizon ascending turn task, as shown in Fig. 5. The optimal solution to this task involves initially performing a horizontal turn, followed by a straight-line acceleration, and

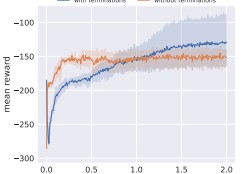 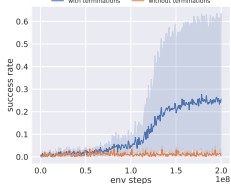 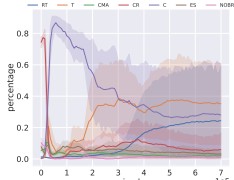 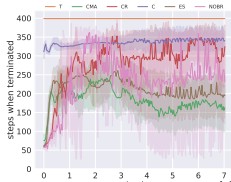

(a) Cumulative reward dur-
ing training

(b) Success rate during
training

(c) Triggering proportions
of each termination condi-
tion during training

(d) Episode length when
termination condition trig-
gers

Figure 6: The left two figures: Comparison between training process with and without termination conditions. The right two figures: Analysis of training process when employing the 7 termination conditions. Results come from experiments over 5 random seeds.

finally ascending in altitude. We sequentially assign these three sub-goals to the aforementioned trained policy, and it can be observed that the policy is able to complete the task effectively. This demonstrates that VVC-Gym supports the modification of the goal for the current episode as necessary (Bøhn et al., 2023). From the inverse perspective, if presented with a complex long-horizon task, VVC-Gym can facilitate research on sub-goal generation methods.

**Investigating demonstration-based RL.** Demonstration-based RL can help overcome the exploration challenge (Ramírez et al., 2022; Gong et al., 2024b). To showcase the suitability of VVC-Gym for studying demonstration-based RL, we conduct experiments on VVC-Gym with the GCBC and RLfD algorithm GCBC+PPO (Vinyals et al., 2019; Baker et al., 2022), with the results presented in Table 3. It is evident that, firstly, the performance of GCBC trained on the

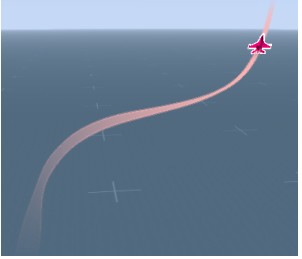

Figure 5: A long-horizon ascending turn task finished by a GCRL policy with pre-defined sub-goals.

relatively low-quality and low-quantity $\overline{\mathcal{D}_E^0}$ far exceeds that of the non-demonstration-based RL algorithms such as SAC, SAC+HER, and PPO as shown in Table 2a. Secondly, both GCBC and GCBC+PPO exhibit improved policy performance as the quantity and quality of the demonstrations increase. This suggests that VVC-Gym and the accompanying demonstrations are well-suited for studying demonstration-based RL, including the investigation of the impact of demonstration quantity and quality on training, and the development of high-performing policies even with imperfect demonstrations.

## 4.4 ABLATION STUDIES

In this section, we conduct ablation studies on the termination conditions and the reward function, examining how their settings affect training. In the experiments, all configurations of the environment and algorithm, except for those being studied in the ablation study, are listed in Appendix F in G.

### 4.4.1 ABLATION ON TERMINATION CONDITIONS

To demonstrate the role of each termination condition during RL training, we conduct ablation studies on termination conditions by training policies using PPO, with experimental details listed in Appendix K. Fig. 6 shows the corresponding results. Fig. 6a and 6b illustrate the trends in cumulative reward and success rate with and without the use of termination conditions. Without them, the policy experiences a rapid rise in cumulative reward, which then plateaus at a certain level, while the success rate remains near zero throughout the training process. The rapid increase in cumulative reward is primarily due to the fact that the reward for triggering T is higher than that for other non-RT termination conditions. Conversely, with the use of termination conditions, the policy's cumulative reward can be consistently enhanced, incrementally raising the success rate.

Fig. 6c and 6d illustrate why the use of termination conditions results in better policies. Fig. 6c shows how the frequency of termination conditions changes over training, while Fig. 6d shows the average length of episodes when termination conditions are triggered. It is evident that employing

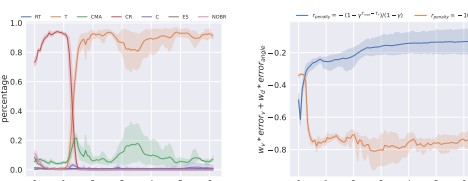

(a) Triggering proportions (b) Weighted sum of errors of each termination condi- in magnitude and direction tion during training of velocity vector

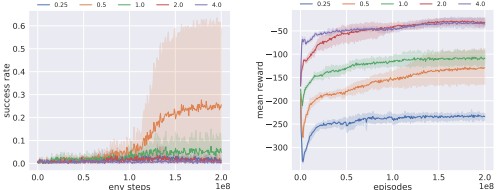

(a) Success rate during (b) Cumulative reward dur- training ing training

Figure 7: Results of ablation on $r_{penalty}$. Results come from experiments over 5 random seeds.

Figure 8: Results of ablation study on $b$. Results come from experiments over 5 random seeds.

termination conditions can lead the training process to exhibit several distinct stages, each with unique characteristics, as outlined in Appendix K. The training stages indicate two key advantages of using termination conditions: First, they enable early termination of unreasonable episodes at various stages, avoiding the collection of meaningless samples after the occurrence of severe states (as shown by Fig. 6d). This results in more episodes sampled within the same computational budget, thus enhancing training efficiency. Second, the training process features several smooth transitions, which aid in the gradual mastery of goal achievement (as shown by Fig. 6c). This incremental learning process eases policy learning, allowing capabilities to evolve progressively.

### 4.4.2 ABLATION ON REWARD FUNCTIONS

In this section, we conduct ablation studies on the settings of $r_{penalty}$, $b$ to analyze the impact of different settings on training. The specific experimental settings are detailed in Appendix L.

**Ablation study on the setting of $r_{penalty}$.** In our experiments, we set the penalty reward $r_{penalty}$ with a large negative constant and a pessimistic estimation, with the results shown in Fig. 7. Fig. 7a illustrates the trend in the proportion of termination conditions triggered during training when using the large negative constant. It is obvious that the agent prefers timeout terminations over other types, indicating a distinct intermediate learning phase where it shifts from frequently triggering non-timeout terminations to timeout termination. Fig. 7b shows the trend in the error between the end-of-episode state and the goal during training. It is observed that the policy intentionally increases the error, or moves away from the goal, to trigger timeout termination and thus avoid the large penalty associated with triggering C, CR, NOBR, ES, and CMA. Furthermore, in Appendix L.2, we provide an analysis of cumulative rewards and success rates, which further corroborates the aforementioned analysis. In summary, employing the large negative constant method introduces an intermediate learning stage where the policy adopts an opportunistic and easier-to-learn manner to trigger timeout terminations by moving away from the goal. This intermediate phase is counterproductive to the agent's goal achievement learning process.

**Ablation study on the setting of $b$.** To analyze the impact of the scaling factor $b$ on policy training, we conduct training with different $b$ and show the corresponding results in Fig. 8. Fig. 8a and 8b show the success rate and cumulative reward during training, respectively. It can be observed that, except for $b = 0.5$, policies trained with other $b$ barely possess the capability to achieve goals. However, the larger the value of $b$, the higher the cumulative reward, indicating that with a larger $b$, the policy tends to have a higher average reward per step. This suggests that with a larger $b$, although the policy may not meet the precision requirements for triggering RT, it can quickly move closer to the goal from a greater distance. However, the low success rate of larger $b$ indicates that it fails in helping the policy to continuously approach the goal. We provide a detailed analysis of the impact of $b$ on velocity direction in the appendix L.3. In summary, the selection of $b$ should be carefully considered based on the task requirements, trading off between the speed and accuracy of achieving goals.

In summary, through ablation studies, we demonstrate that both termination conditions and reward structure have a significant impact on GCRL training, affecting both the training process and the final training outcomes. The default termination conditions and general reward structure of VVC-Gym facilitate researchers in exploring the effects of various environment designs on GCRL training.

## 5 CONCLUSION, LIMITATIONS, AND FUTURE WORK

In this paper, we introduce VVC-Gym, a fixed-wing UAV environment tailored for researching multi-goal long-horizon problems, accompanied by multi-quality demonstration sets. We offer a comprehensive description of the VVC-Gym design and the methodology for generating demonstrations. Additionally, we provide several baselines for Goal-Conditioned Reinforcement Learning (GCRL) and demonstration-based Reinforcement Learning (RL) on VVC-Gym and the accompanying demonstrations. Finally, we conduct ablation studies on environment designs, illustrating that VVC-Gym is well-suited for investigating the impact of environment designs on GCRL training.

Despite the contributions of our work, there are several limitations and directions for extension that should be noted in future research: (1) construct tasks with longer control sequences, including Basic Flight Maneuvers (BFMs) such as Slow Roll and Knife Edge, to facilitate GCRL researchers in utilizing VVC-Gym as a challenging long-horizon task environment for their studies. (2) Establish baselines for automatic sub-goal generation methods and offline RL algorithms (Liu et al., 2024; 2025). (3) Explore methods for collecting low-cost demonstrations for velocity vector control tasks from human play data.

ACKNOWLEDGMENTS

This work was supported by the Science and Technology Innovation Program of Hunan Province (No. 2023RC1005) and partly supported by the National Natural Science Foundation of China (No. 62306325).

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

# Appendix

## A EQUATIONS OF MOTION

The Equations of Motion (EoM) of a Fixed-Wing aircraft is comprised of translational and rotational nonlinear equations of a 6-Degrees-of-Freedom (6-DoF) system about the 3 Body Coordinate System (BCS) axes. This generates a 12-dimensional fully observable state space, positions and their rates Clarke & Hwang (2020): the velocity component along the OX axis of the BCS $u$, the velocity component along the OY axis of the BCS $v$, the velocity component along the OZ axis of the BCS $w$, the roll angle $\phi$, the pitch angle $\theta$, the yaw angle $\psi$, the roll rate $p$, the pitch rate $q$, the yaw rate $r$, the coordinate of the x-axis in the Geodetic Coordinate System (GCS) $x_E$, the coordinate of the y-axis in the GCS $y_E$, and the coordinate of the z-axis in the GCS $z_E$.

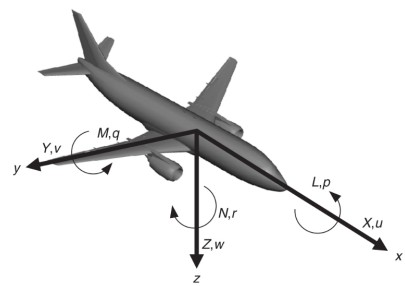

Figure 9: Body coordinate system of fixed-wing aircraft (from Fig.3.9 of Allerton (2009)).

$$\frac{du}{dt} = \frac{T_x - D\cos\alpha\cos\beta - Y\cos\alpha\sin\beta + L\sin\alpha - mg\sin\theta}{m} + rv - qw \tag{2}$$

$$\frac{dv}{dt} = \frac{T_y - D\sin\beta + Y\cos\beta + mg\sin\phi\cos\theta}{m} + pw - ru \tag{3}$$

$$\frac{dw}{dt} = \frac{T_z - D\sin\alpha\cos\beta - Y\sin\alpha\sin\beta - L\cos\alpha + mg\cos\phi\cos\theta}{m} + qu - pv \tag{4}$$

$$\frac{d\phi}{dt} = p + \tan\theta\left(q\sin\phi + r\cos\phi\right) \tag{5}$$

$$\frac{d\theta}{dt} = q\cos\phi - r\sin\phi \tag{6}$$

$$\frac{d\psi}{dt} = \frac{1}{\cos\theta}\left(q\sin\phi + r\cos\phi\right) \tag{7}$$

$$\frac{dp}{dt} = \left(c_1 r + c_2 p\right)q + c_3 M_x + c_4 M_z \tag{8}$$

$$\frac{dq}{dt} = c_5 pr - c_6\left(p^2 - r^2\right) + c_7 M_y \tag{9}$$

$$\frac{dr}{dt} = \left(c_8 p - c_2 r\right)q + c_4 M_x + c_9 M_z \tag{10}$$

where $T$ is the thrust force, $D$ is the drag force, $Y$ is the side force, $L$ is the lift force, $M$ is the moment ($T, D, Y, L, M$ are calculated by an aerodynamic model with open-source aircraft model Simulator (2022). Please refer to the Fig.1 of Wang & Wang (2020) for the detailed calculation process.), $m$ is the mass of the aircraft, $g$ is the gravitational acceleration, $c_1 = \frac{(I_y - I_z)I_z - I_{zx}^2}{I_x I_z - I_{zx}^2}$, $c_2 = \frac{(I_x - I_y + I_z)I_{zx}}{I_x I_z - I_{zx}^2}$, $c_3 = \frac{I_z}{I_x I_z - I_{zx}^2}$, $c_4 = \frac{I_{zx}}{I_x I_z - I_{zx}^2}$, $c_5 = \frac{I_z - I_x}{I_y}$, $c_6 = \frac{I_{zx}}{I_y}$, $c_7 = \frac{1}{I_y}$, $c_8 = \frac{(I_x - I_y)I_x + I_{zx}^2}{I_x I_z - I_{zx}^2}$, and $c_9 = \frac{I_x}{I_x I_z - I_{zx}^2}$, where $I_x, I_y, I_z, I_{zx}$ are moment of inertia.

Table 4: Comparison of efficiency between VVC-Gym and popular RL environments. The content within the parentheses indicates the physical engine used in the environment.

|  | Reach (Panda) | HalfCheetah (Mujoco) | Ant (Mujoco) | Hammer (Adroit) | VVC-Gym |
|---|---|---|---|---|---|
| FPS | 2189 | 4042 | 2504 | 2308 | 3623 |
| Wall-clock time (s) | 4506 | 2435 | 3947 | 4306 | 2854 |

For the linear velocities along the three axes in the Global Coordinate System (GCS), denoted as $\nu_x, \nu_y, \nu_z$, the relationships are as follows

$$\nu_x = \frac{dx_E}{dt} = u\cos\theta\cos\psi + v(\sin\theta\sin\phi\cos\psi - \cos\phi\cos\psi) + w(\sin\theta\cos\phi\cos\psi + \sin\phi\sin\psi) \tag{11}$$

$$\nu_y = \frac{dy_E}{dt} = u\cos\theta\sin\psi + v(\sin\theta\sin\phi\sin\psi + \cos\phi\cos\psi) + w(\sin\theta\cos\phi\sin\psi - \sin\phi\cos\psi) \tag{12}$$

$$\nu_z = \frac{dz_E}{dt} = -u\sin\theta + v\sin\phi\cos\theta + w\cos\phi\sin\theta \tag{13}$$

Besides, the airspeed $V = \sqrt{\nu_x^2 + \nu_y^2 + \nu_z^2} = \sqrt{u^2 + v^2 + w^2}$, flight path elevator angle $\mu = \arcsin(\frac{\nu_z}{V})$, the flight path azimuth angle $\chi = \arctan(\frac{\nu_y}{\nu_x})$, angle of attack $\alpha = \arctan(\frac{w}{u})$, and sideslip angle $\beta = \arcsin(\frac{\nu}{V})$.

Although VVC-Gym is implemented in Python, we implement the aforementioned computations with C++, which renders the simulations in VVC-Gym highly efficient. On a single node (equipped with an Intel Core i9-10980XE CPU, NVIDIA GeForce RTX 3090, and 128GB of memory), we train for $10^7$ steps using the PPO algorithm (with 64 rollout workers) within the StableBaselines3 framework and compare VVC-Gym with commonly used RL environments. The results are presented in Table 4. It can be observed that VVC-Gym achieves the sampling speed of environments commonly used in academic research and even surpasses several of them.

## B  THE CONTROL LAW

The control law is responsible for converting the pilot's commands or the signals from the automatic flight system into precise movements of the aircraft's control surfaces, such as the ailerons, elevators, and rudder, to achieve stable flight and precise maneuvering of the aircraft. Our control law model receives overload command $n_{zc}$ and roll rate command $p_c$ as input, and outputs the deflections of the elevator and aileron, denoted as $\delta_e$ and $\delta_a$, respectively,

$$\delta_e = k_{p,e}\left(n_z - n_{zc}\right) + k_{i,e}\int \left(n_z - n_{zc}\right)dt + k_{d,e}q \tag{14}$$

$$\delta_a = k_{p,a}\left(p - p_c\right) + k_{i,a}\int \left(p - p_c\right)dt \tag{15}$$

where $k_{p,e}, k_{i,e}, k_{d,e}$ are the proportional gain, integral gain, and derivative gain of the PID controller that controls the elevator, and $k_{p,a}, k_{i,a}$ are the proportional gain and integral gain of the PID controller that controls the aileron.

## C   THE PID CONTROLLER OF GUIDANCE LAW FOR VELOCITY VECTOR CONTROL

The expert model of the guidance law takes desired velocity vector $v_c, \mu_c, \chi_c$ as input and produces overload command $n_{zc}$, roll rate command $p_c$, and the deflection of the power level actuator $\delta_{pla}$:

$$n_{yc,e} = \frac{\Delta\chi}{T_\chi} \cdot V_{g,hor}/g \tag{16}$$

$$n_{zc,e} = \frac{\Delta\mu}{T_\mu} \cdot V_g/g + \cos\mu \tag{17}$$

$$n_{zc} = \sqrt{n_{yc,e}^2 + n_{zc,e}^2} \tag{18}$$

$$\phi_c = \tan^{-1}\left(\frac{n_{yc,e}}{\cos\theta}\right) \tag{19}$$

$$p_c = k_{p,p}\left(\phi_c - \phi\right) + k_{d,p}p \tag{20}$$

$$\delta_{pla} = k_{p,pla}\left(v_c - v\right) + \Delta_{pla} \tag{21}$$

where $n_{yc,e}$ is the lateral overload command in GCS, $n_{zc,e}$ is the normal overload command in GCS, $\Delta\mu = \mu_c - \mu$ is the error in flight path elevator angle, $\Delta\chi = \chi_c - \chi$ is the error in flight path azimuth angle, $T_\chi$ is the estimation of the time required for the aircraft to reach the desired flight path azimuth angle, $T_\mu$ is the estimation of the time required for the aircraft to reach the desired flight path elevator angle, $V_g$ is the aircraft ground speed, $V_{g,hor}$ is the horizontal component of $V_g$, $\phi_c$ is an intermediate variable representing the roll angle command, $k_{p,p}, k_{d,p}$ are the proportional gain and integral gain of the PID controller that controls the roll rate command, $k_{p,pla}$ is the proportional gain of the PID controller that controls the deflection of the power level actuator, and $\Delta_{pla}$ is constant that ensures that the aircraft maintains its velocity without decrease during level flight.

## D   GENERATING DEMONSTRATIONS

**Discretizing the goal space.** The goal space $G = [v_{min}, v_{max}] \times [\mu_{min}, \mu_{max}] \times [\chi_{min}, \chi_{max}]$ is discretized to $G' = \{v_{min} + i\frac{v_{max}-v_{min}}{N_v-1} | i \in [1, 2, \ldots, N_v - 1]\} \times \{\mu_{min} + i\frac{\mu_{max}-\mu_{min}}{N_\mu-1} | i \in [1, 2, \ldots, N_\mu - 1]\} \times \{\chi_{min} + i\frac{\chi_{max}-\chi_{min}}{N_\chi-1} | i \in [1, 2, \ldots, N_\chi - 1]\}$, where $N_v, N_\mu, N_\chi$ are the number that we equally divide the $v, \mu, \chi$ spaces. In the experiments, we utilize the parameters specified in Table 5 to generate the discrete goal set.

Table 5: Parameters used to discretize the goal space.

|     | $v$ | $\mu$ | $\chi$ |
|-----|-----|-------|--------|
| min | 100 | -85   | -170   |
| max | 300 | 85    | 170    |
| N   | 21  | 35    | 69     |
| #discretized goals: 50715 | | | |

**Augmenting demonstrations based on symmetry.** If $\mathcal{D}_E$ contain a trajectory $\tau_g = \{(s_1, a_1), \ldots, (s_T, a_T)\}$ that achieves goal $g = (v, \mu, \chi)$, then $\mathcal{D}_E$ is augmented with $\tau_{g^{-1}} = \{(s_1^{-1}, a_1^{-1}), \ldots, (s_T^{-1}, a_T^{-1})\}$, where $g^{-1} = (v, \mu, -\chi), s^{-1} = (\theta, -\phi, -\psi, -\chi, \mu, h, -p, v), a^{-1} = (-p, n_z, \delta_{pla})$. If $\mathcal{D}_E$ has a trajectory that can achieve $g^{-1}$, then the shorter one is kept.

## E   ALGORITHM BACKGROUND

### E.1   GOAL-CONDITIONED REINFORCEMENT LEARNING

Let $\Delta(\mathcal{X})$ denote the probability distribution over a set $\mathcal{X}$. Goal-conditioned RL is described by goal-augmented MDP Liu et al. (2022); Pitis et al. (2020) $M = \langle \mathcal{S}, \mathcal{A}, \mathcal{T}, r, \gamma, \mathcal{G}, p_g, \zeta \rangle$, where $\mathcal{S}, \mathcal{A}, \gamma$ are the state set, action set, and discount factor, $\mathcal{T} : \mathcal{S} \times \mathcal{A} \rightarrow \Delta(\mathcal{S})$ is the transition probabilities, $r = \{r_g | r_g : \mathcal{S} \rightarrow \mathbb{R}, g \in G\}$ is the goal-conditioned reward functions, $\mathcal{G}$ is the space of goals, $p_g$ is the desired goal distribution, and $\zeta : \mathcal{S} \rightarrow \mathcal{G}$ is a tractable mapping function that maps the state to a specific goal. For a fixed goal $g$, solving it means finding optimal policy from a standard MDP $M_g = \langle \mathcal{S}, \mathcal{A}, \mathcal{T}, r_g, \gamma \rangle$. There are three kinds of goals in goal-conditioned RL: desired goal is the requirements of the task, and the desired goal distribution is denoted as $p_{dg}$; achieved goal is the corresponding goal achieved by the current timestamp and state, and the

Table 6: Parameters used in Environment

(a) Normal Version

| Parameter | Value |
|---|---|
| $v_{min}, v_{max}$ | **150, 250** |
| $\mu_{min}, \mu_{max}$ | **-10, 10** |
| $\chi_{min}, \chi_{max}$ | **-30, 30** |
| simulation frequency | 10 |
| $T_{max}$ | 400 |
| $w_v, w_d$ | 0.5, 0.5 |
| $\sigma_v, \sigma_d$ | 100, 180 |
| $b$ | 0.5 |
| $r_{penalty}$ | pessimistic estimation |
| $T_R, T_M$ | 10, 20 |
| $h_0$ | 0 |
| $\phi_{max}$ | 720 |
| $v_{max}, p_{max}$ in ES | 400, 300 |
| $\phi_N, T_N$ | 60, 20 |

(b) Hard Version

| Parameter | Value |
|---|---|
| $v_{min}, v_{max}$ | **100, 300** |
| $\mu_{min}, \mu_{max}$ | **-85, 85** |
| $\chi_{min}, \chi_{max}$ | **-170, 170** |
| simulation frequency | 10 |
| $T_{max}$ | 400 |
| $w_v, w_d$ | 0.5, 0.5 |
| $\sigma_v, \sigma_d$ | 100, 180 |
| $b$ | 0.5 |
| $r_{penalty}$ | pessimistic estimation |
| $T_R, T_M$ | 10, 20 |
| $h_0$ | 0 |
| $\phi_{max}$ | 720 |
| $v_{max}, p_{max}$ in ES | 400, 300 |
| $\phi_N, T_N$ | 60, 20 |

achieved goal distribution is denoted as $p_{ag}$; behavioral goal is the goal for sampling trajectory in the current episode Liu et al. (2022). The policy parameterized by $\theta$ is modeled as $\pi_\theta : \mathcal{S} \times \mathcal{G} \to \Delta(\mathcal{A})$ based on the idea of universal value function approximators (UVFA) Schaul et al. (2015). The objective of goal-conditioned RL is to maximize the expectation $\mathbb{E}_{g \sim p_{dg}, \pi}[\sum_{t=0}^{\infty} \gamma^t r_g(s_t)]$.

### E.2 IMITATION LEARNING

Imitation learning is a demonstration-driven, sample-efficient method for learning policies by imitating demonstrators Belkhale et al. (2023). IL assumes access to a dataset $\mathcal{D}_E = (\tau_1, \ldots, \tau_N)$ of $N$ demonstrations. $\tau_i = \{(s_1, a_1), \ldots, (s_{T_i}, a_{T_i})\}$ is a sequence of length $T_i$ of state-action pairs sampled by the demonstrator $\pi_E(\cdot|s_t)$ through environment dynamics $\mathcal{T}(\cdot|s_t, a_t)$. The objective of IL is to learn a policy $\pi_\theta : \mathcal{S} \to \Delta(\mathcal{A})$ parameterized by $\theta$ from $\mathcal{D}_E$. Behavioral cloning Pomerleau (1991) is a widely applied IL method, which learns the imitation policy by optimizing a supervised loss to maximize the likelihood of demonstrator actions Sasaki & Yamashina (2020):

$$\mathcal{L}(\theta) = -\mathbb{E}_{(s,a) \sim \mathcal{D}_E}[\log \pi_\theta(a|s)]. \tag{22}$$

## F TWO VERSIONS OF VELOCITY VECTOR CONTROL TASKS

We implement two versions of the velocity vector control task in VVC-Gym: the normal version and the hard version, which are detailed in Table 6. The distinction between the two versions lies in the definition of the goal space. The hard version features a larger goal space, which encompasses nearly the entire three-dimensional space, making it more challenging.

## G ALGORITHM DETAILS

The Imitation Gleave et al. (2022) framework is utilized to implement behavioral cloning (BC) Pomerleau (1991) algorithm with parameters listed in Table 7a, and the Stable Baselines3 Raffin et al. (2021) framework for SAC and PPO with parameters listed in Table 7b and 7c. 128*128 fully connected network and the Tanh activation function are used.

## H SUPPORT FOR RESEARCH ON NON-MARKOVIAN REWARD PROBLEMS

The values of $T_R, T_M, T_N$ in termination conditions R, CMA, and NOBR respectively shape the reward function's characteristics, impacting the policy's form. When $T_R = T_M = T_N = 1$, rewards are Markovian that only depend on the current state, allowing policies to be based solely on the current state. If any of $T_R, T_M, T_N$ exceed 1, rewards become dependent on a history of consecutive

Table 7: Algorithm configurations.

(a) BC

| Parameter | Value |
|-----------|-------|
| l2_weight | 0 |
| ent_weight | $10^{-2}$ |
| batch_size | 4096 |
| epochs | 300 |

(b) SAC

| Parameter | Value |
|-----------|-------|
| ent_coef | 'auto' |
| gamma | 0.995 |
| lr | $3 \times 10^{-4}$ |
| batch_size | 1024 |
| buffer_size | $2 \times 10^5$ |
| learning_starts | 10240 |
| gradient_steps | 1 |
| train_steps | $10^6$ |
| rollout_process_num | 1 |
| use_sde | False |
| HER:n_sampled_goal | 4 |

(c) PPO

| Parameter | Value |
|-----------|-------|
| ent_coef | $10^{-2}$ |
| gamma | 0.995 |
| gae_lambda | 0.95 |
| lr | $3 \times 10^{-4}$ |
| batch_size | 4096 |
| train_steps | $5 \times 10^8$ |
| rollout_process_num | 64 |
| n_steps | 2048 |
| n_epochs | 5 |
| use_sde | True |
| normalize_advantage | True |

Table 8: Success rate on MR and various NMR problems. Results come from experiments over 5 random seeds.

| $T_R$ | Reward type | Success rate |
|-------|-------------|--------------|
| 1 | MR | 84.43±4.74 |
| 10 | NMR | 65.53±12.95 |
| 20 | NMR | 45.03±10.65 |
| 30 | NMR | 21.99±6.63 |

states, making them non-Markovian, and policies must account for state history Abel et al. (2021); Gaon & Brafman (2020); Abel et al. (2022).

**Investigating non-Markovian reward problems.** Non-Markovian reward (NMR) refers to the reward that depends on multiple steps of states and action Abel et al. (2021), such as determining whether an UAV has stably achieved a desired velocity vector based on a sequence of states. Typically, the longer the state sequence that a NMR depends on, the more challenging it becomes to train policies with RL. To illustrate the impact of NMR on RL training, we set different values of $T_R$ in VVC-Gym, which represents the dependency of the NMR on the length of the state sequence. The results are presented in Table 8. It is observed that as $T_R$ increases, the policy's performance deteriorates. Additionally, adjusting $T_M$ and $T_N$ can also alter the Markovian nature of reward function. This indicates that VVC-Gym is well-suited for studying NMR problems.

## I  THE TWO CONTROL MODES OF VVC-GYM

We train both a guidance law model and an end-to-end model using SAC on the easy version of VVC-Gym, and present the corresponding results in Table 9. It can be observed that the trained guidance law model demonstrates superior performance. The reason for this is that the control law model provided in Appendix B is capable of stabilizing the control outputs, which serves a purpose analogous to the temporal abstraction achieved through frame-skipping, thereby reducing the temporal complexity of exploration.

The above results can be interpreted from two perspectives: On one hand, the use of a simple control law model to stabilize RL outputs helps address the temporal complexity of exploration. On the other hand, although the end-to-end mode is more difficult to train, it also serves as a more challenging testbed for evaluating RL exploration methods. Additionally, the end-to-end mode can also be employed to investigate the oscillation action problem Mysore et al. (2021) in RL.

Table 9: Success rate of SAC between training a guidance law model and training a end-to-end model. Results come from experiments over 5 random seeds.

|  | Guidance law model | End-to-end model |
|---|---|---|
| Success Rate (%) | 90.40±0.49 | 7.60±5.08 |

Table 10: Success rate (%) of GCRL algorithms on Reach, PointMaze, and VVC. The demonstrations used in the Reach experiments are from the official script provided by Panda-Gym, the demonstrations used in the PointMaze experiments are from Minari, and the demonstrations used in the VVC experiments are $\mathcal{D}_E^0$. Results come from experiments over 5 random seeds.

|  | MEGA | GCBC | GCBC+PPO |
|---|---|---|---|
| Reach | 100.0±0.0 | 70.63±2.99 | 100.0±0.0 |
| PointMaze | 100.0±0.0 | 75.96±5.34 | 93.33±3.06 |
| VVC | 8.32±1.86 | 17.08±0.57 | 38.31±1.62 |

## J  CHALLENGE OF THE VVC TASK

Fixed-wing UAV's VVC is a challenging task. The difficulties lie in: (1) The large exploration space of the policy, which is a continuous state, continuous action problem, and the policy requires additional exploration of the goal space during training. (2) The long interaction sequences, with the average length of demonstrations exceeding 280. Even well-trained policies require an average of over 100 steps to achieve a goal, and more challenging goals can demand upwards of 300 steps (see Table 1 for corresponding results).

We provide evidence that VVC is a challenging task through the following three sets of experiments:

**Firstly, standard RL algorithms struggle to solve the VVC task**. Table 2a shows the success rates of SAC (1.08%) and PPO (0.04%). It is evident that SAC and PPO struggle to solve the VVC task.

**Secondly, existing GCRL algorithms can effectively solve common multi-goal tasks in academic research, but they can only solve the VVC task to a certain extent**. We compare the performance of different GCRL algorithms on VVC and common multi-goal tasks in academic research, PointMaze (PointMaze_Large_DIVERSE_G-v3) and Reach (PandaReach-v3). The results are shown in Table 10. It can be seen that these GCRL algorithms can almost completely solve Reach and PointMaze, but the best algorithm achieves only a $38.31\%$ success rate on VVC. These results indirectly reflect that VVC is a challenging task.

**Thirdly, the human-designed classical PID controller (detailed in Appendix C) has only a $20.08\%$ success rate**, which also indirectly reflects that VVC is a challenging task.

We believe that for academic research, the difficulty of the task should progress in tandem with the research on algorithms. The task should have appropriate levels of difficulty to properly evaluate different algorithms. If the task is too easy, too hard, or unsolvable, it will fail to provide a useful signal for benchmarking. Therefore, we believe that the current success rates of GCRL algorithms on VVC being less than 50% is helpful for researchers to discover more insights when designing algorithms.

## K  ABLATION ON TERMINATIONS

### K.1  EXPERIMENT DETAILS

For the experiment with termination conditions, all of the 7 termination conditions are used in the environment, with parameters listed in 6a. For the experiment without terminations conditions, only RT, T, and ES are used in the environment, with the corresponding parameters the same as 6a. In the comparative experiments, the primary purpose of using ES is to prevent excessive values of linear velocity $v$ and angular velocity $p$, which could lead to floating-point overflow errors. However, we utilize the largest possible values for $v_{max} = 600, p_{max} = 600$ to ensure that ES operates with the

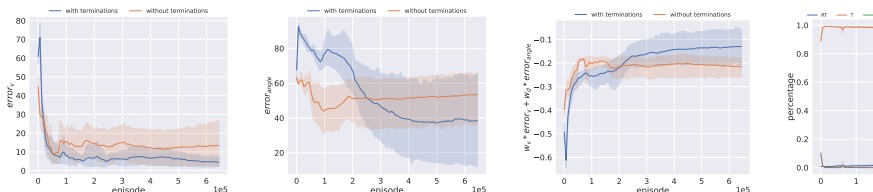

(a) Error in magnitude of velocity vector (b) Error in direction of velocity vector (c) Weighted sum of errors in magnitude and direction of velocity vector (d) Triggering proportions of each termination condition during training

Figure 10: Statistical information on training process of termination conditions. Results come from experiments over 5 random seeds.

minimal necessary intervention. For the PPO algorithm, both experiments use parameters listed in 7c.

### K.2 ADDITIONAL ANALYSIS ON THE ABLATION OF TERMINATION CONDITIONS

From Fig. 6c and 6d, it can be observed that when termination conditions are employed, the entire training process can be divided into four stages:

In the first stage, when the number of training episodes is less than $10^4$, the policy lacks any meaningful capability and tends to explore randomly. The policy frequently triggers CR, NOBR, ES, and CMA, with short average episode lengths when these conditions are triggered.

The second stage occurs when the number of training episodes is between $10^4$ and $5 * 10^4$. During this stage, the frequency of CR being triggered significantly decreases, while the frequency of C being triggered increases substantially. Additionally, the average episode lengths for CR, NOBR, ES, and CMA are notably longer. This indicates that in this stage, the policy has learned how to quickly avoid triggering CR, NOBR, ES, and CMA. And the higher frequency of triggering C is due to the policy's lack of capability to continuously approach the goal, leading to frequent triggering of C.

The third stage is when the number of training episodes is between $5 * 10^4$ and $3 * 10^5$. In this stage, the frequency of C being triggered significantly decreases, while the frequency of T being triggered rapidly increases. This suggests that as training progresses in this stage, the policy begins to acquire the ability to approach the goal, but it is not yet able to do so quickly or continuously, resulting in frequent triggering of T.

The fourth stage occurs when the number of training episodes exceeds $3 * 10^5$. In this stage, the policy's ability to approach the desired goal is gradually improved, leading to an increasing frequency of triggering RT. The policy starts to possess the capability to complete parts of the goals.

Furthermore, the trend in the error between the state at the end of episode and the goal, which is present in Fig. 10, also corroborates the aforementioned four stages exhibited by the learning process when termination conditions are used.

## L ABLATION ON REWARD FUNCTIONS

### L.1 EXPERIMENT DETAILS.

Aside from the parameters investigated in the ablation study, all other environment and algorithm configuration parameters are consistent with those detailed in Table 6 and 7. In the ablation study on $r_{penalty}$, $r_{penalty} = -1000$ is employed for the large negative constant method, and $r_{penalty} = -\frac{1-\gamma^{T_{max}-T_\tau}}{1-\gamma}$ is employed for the pessimistic estimation method, where $T_\tau$ is the length of current sampled episode $\tau$. In the ablation study on the scaling factor $b$, 0.25, 0.5, 1.0, 2.0, and 4.0 are employed for training. From the previous experimental results, it is evident that controlling the

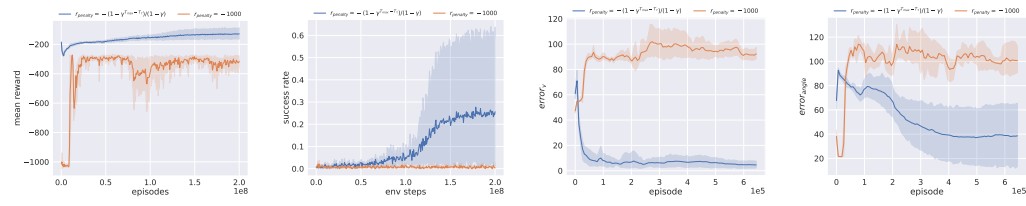

(a) Cumulative reward during training  (b) Success rate during training  (c) Error in magnitude of velocity vector  (d) Error in direction of velocity vector

Figure 11: Statistical information on training process of different $r_{penalty}$. Results come from experiments over 5 random seeds.

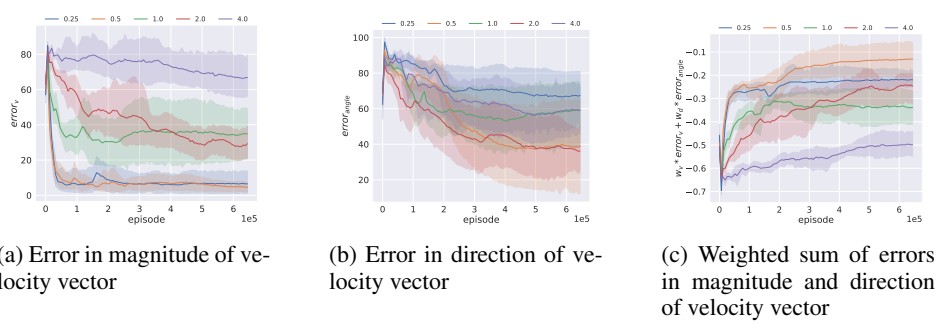

(a) Error in magnitude of velocity vector  (b) Error in direction of velocity vector  (c) Weighted sum of errors in magnitude and direction of velocity vector

Figure 12: Errors of velocity vector of policies trained with different $b$ during training. Results come from experiments over 5 random seeds.

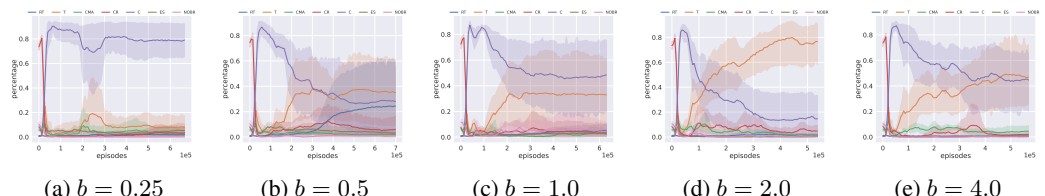

(a) $b = 0.25$  (b) $b = 0.5$  (c) $b = 1.0$  (d) $b = 2.0$  (e) $b = 4.0$

Figure 13: Triggering proportions of each termination condition during training. Results come from experiments over 5 random seeds.

magnitude of velocity is simpler than controlling the direction of velocity. Therefore, we used a larger $w_d$ compared to $w_v$, specifically setting $w_d$ to 0.5, 0.625, 0.75, and 0.875 for training.

## L.2 ADDITIONAL ANALYSIS OF THE ABLATION ON $r_{penalty}$.

Fig. 11c and Fig. 11d respectively depict the error in magnitude of velocity vector and the error in direction of velocity vector during training. It is evident that when a large negative constant is employed, the policy opportunistically moves away from the goal to trigger the timeout termination, thereby avoiding the greater punishment associated with non-timeout termination. The above analysis aligns with our conclusion in Sec. 4.4.2.

Additionally, Fig. 11a and 11b illustrate the changes in cumulative reward and success rate during training for the two methods of setting $r_{penalty}$. When using the large negative constant method, the cumulative reward plateaus at around -300 after rising from -1000, and the policy's success rate remains nearly zero throughout, indicating that the policy has learned to avoid triggering C, CR, NOBR, ES, and CMA by frequently triggering timeout terminations, and it is difficult to learn to achieve goals based on this behavior.

Table 11: Error in the direction of the velocity vector and the immediate reward at the last step of the trajectory finished by policies trained with different $b$. The 'Error interval' indicates the range to which the error in the direction of the velocity vector must decrease before the reward change rate exceeds 1 (The curve of reward varying with error can be found in Fig. 2). The 'Error between the state at the end of episode and the goal' represents the error in the direction of the velocity vector between the last state of the trajectory finished by policies and the goal. The 'Immediate reward at the last step of episode' is the immediate reward of the last state of the trajectory finished by policies.

| $b$ | Error interval (reward change rate ¿ 1) | Error between the state at the end of episode and the goal | Immediate reward at the last step of episode |
|---|---|---|---|
| 0.25 | (0, 63.64) | 70 | -0.79 |
| 0.5 | (0, 45) | 40 | -0.47 |
| 1.0 | / | 60 | -0.33 |
| 2.0 | (90, 180) | 40 | -0.05 |
| 4.0 | (113.39, 180) | 60 | -0.01 |

L.3    ADDITIONAL ANALYSIS OF THE ABLATION ON $b$.

Fig. 12a and 12b present the trends in the error of velocity magnitude and velocity direction between the state at the end of an episode and the goal during training. Controlling the velocity magnitude of the UAV is relatively simple, and a smaller $b$ is more beneficial for the UAV to improve the accuracy of reaching the desired velocity magnitude. Controlling the velocity direction of the UAV is more challenging, and the experiments with $b = 0.5$ and $b = 2.0$ achieved the best results. This is because when $b > 1$, the reward function changes rapidly in the region of high error but slowly in the region of low error, which is beneficial for the policy to quickly approach the goal when the error is large but not beneficial for the policy to continuously approach the goal when the error is small; when $b < 1$, the reward function's characteristics are opposite. Therefore, a $b$ slightly greater or less than 1 can effectively balance the policy's speed in different error intervals, somewhat improving the precision of approaching the goal, as shown in Fig. 12c. However, when $b$ is too large or too small, it significantly increases the policy's speed in one interval but severely reduces it in another, which, as evidenced by the experimental results, is not beneficial for the policy to achieve goals.

Fig. 13 presents the proportion of termination conditions triggered by policies trained with different $b$ during training. From an overall perspective, as learning progresses, the policy effectively learns to avoid triggering the four termination conditions CMA, CR, ES, and NOBR, with C and T dominating the training process for most of the time; (2) as $b$ increases, the proportion of policy triggering T also increases, indicating that a larger $b$ is beneficial for the policy to approach the goal. This is consistent with the analysis of Fig. 12.

Additionally, Table 11 presents the immediate reward at the last step of the completed trajectory and the error between the state and the goal. It is observed that only when $b = 0.5$ can the policy reduce the error to a region where the reward change rate exceeds 1. When $b$ is too small, the reward remains in the error range where $\frac{dr}{de} < 1$, which is not conducive to the policy quickly approaching the goal. When $b = 0.25$, the policy does not even reduce the error to the $\frac{dr}{de} > 1$ error range. When $b$ is too large, the reward remains in the error range where $\frac{dr}{de} < 1$, which is not conducive to the policy continuously approaching the goal. When $b = 2$ and $b = 4$, although the policy can reduce the error to the $\frac{dr}{de} < 1$ error range, the slow change in reward with error makes it difficult for the policy to learn, thus preventing the policy from continuously reducing the error to trigger RT.

In summary, When $b < 1$, it is beneficial for the policy to learn the ability to continuously approach the goal in the error-small interval, but it is not conducive to the policy acquiring the ability to quickly approach the goal in the error-large interval. Conversely, when $b > 1$, it is beneficial for the policy to acquire the ability to quickly approach the goal in the error-large interval, but it is not conducive to the policy acquiring the ability to continuously approach the goal in the error-small interval. Therefore, the selection of $b$ should be carefully considered based on the task requirements, balancing the trade-off between the speed and accuracy of achieving goals.

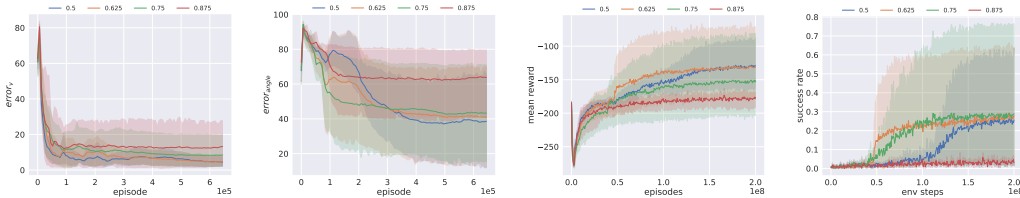

(a) Errors in magnitude of velocity vector (b) Errors in direction of velocity vector (c) Cumulative reward during training (d) Success rate during training

Figure 14: Results of ablation study on $w_v, w_d$. Results come from experiments over 5 random seeds.

Table 12: The Smoothness Measure of State and Action of Each Demonstration. As there is almost no difference between $\mathcal{D}_E^i$ and $\overline{\mathcal{D}_E^i}, i \in \{0, 1, 2, 3\}$, we show the smoothness measure of $\overline{\mathcal{D}_E^i}$ in this table. Optimal values are highlighted in bold, and sub-optimal values are underlined.

| Demonstration | | state | | | | | | | | action | | |
|---|---|---|---|---|---|---|---|---|---|---|---|---|
| | | $\theta$ | $\phi$ | $\psi$ | $\chi$ | $\mu$ | $h$ | $p$ | $v$ | $p$ | $n_z$ | $\delta_{pla}$ |
| $\overline{\mathcal{D}_E^0}$ | $s_1$ | 0.25±0.05 | **0.06±0.03** | **0.23±0.12** | **0.23±0.12** | **0.04±0.05** | **1.47±1.31** | **0.64±0.47** | **0.15±0.09** | **0.46±0.23** | **0.006±0.005** | **0.0015±0.0016** |
| | $s_2$ | 0.14±0.09 | 1.93±2.14 | 0.36±0.16 | 0.31±0.16 | 0.08±0.06 | 2.24±1.64 | 30.65±45.14 | 0.20±0.12 | 2.11±2.21 | 0.03±0.07 | 0.0028±0.0026 |
| $\overline{\mathcal{D}_E^1}$ | $s_1$ | **0.20±0.14** | 0.31±0.49 | 0.57±0.31 | 0.56±0.28 | 0.19±0.14 | 4.46±3.08 | 1.35±0.77 | 0.31±0.17 | 1.44±0.64 | 0.05±0.03 | 0.0044±0.0035 |
| | $s_2$ | 0.81±0.44 | 3.18±2.17 | 1.16±0.87 | 0.81±0.51 | 0.45±0.21 | 6.34±3.90 | 16.00±23.16 | 0.47±0.19 | 7.54±6.51 | 0.47±0.54 | 0.016±0.018 |
| $\overline{\mathcal{D}_E^2}$ | $s_1$ | 0.25±0.18 | 0.31±0.39 | 0.60±0.37 | 0.58±0.31 | 0.24±0.18 | 4.83±3.18 | 1.43±0.83 | 0.31±0.17 | 1.52±0.92 | 0.05±0.03 | 0.0043±0.0036 |
| | $s_2$ | 0.82±0.38 | 3.34±2.02 | 1.30±1.07 | 0.83±0.46 | 0.52±0.23 | 6.95±3.94 | 18.55±26.42 | 0.48±0.20 | 10.98±11.65 | 0.43±0.47 | 0.016±0.015 |
| $\overline{\mathcal{D}_E^3}$ | $s_1$ | 0.28±0.20 | 0.39±0.65 | 0.60±0.39 | 0.58±0.31 | 0.27±0.18 | 4.83±3.11 | 1.51±0.93 | 0.31±0.18 | 1.55±0.99 | 0.06±0.03 | 0.0042±0.0038 |
| | $s_2$ | 0.82±0.35 | 3.34±2.61 | 1.26±1.00 | 0.81±0.39 | 0.54±0.23 | 7.15±3.81 | 20.15±27.62 | 0.49±0.19 | 11.94±13.64 | 0.48±0.48 | 0.018±0.021 |

### L.4 ABLATION STUDY ON THE SETTING OF $w_v$ AND $w_d$.

In the experimental process, we discovered that Eq. 1 can be further refined by breaking down the term $-(\frac{\|\zeta(s_t)-g\|}{\sigma})^b$ into two separate components: $-(w_v(\frac{\|\vec{v}_t-\vec{v}_g\|_t}{\sigma_v})^b + w_d(\frac{\|\vec{v}_t-\vec{v}_g\|_d}{\sigma_d})^b)$, where $\|.\|_t$ calculates the difference in magnitude of two velocity vectors, and $\|.\|_d$ calculates the difference in the direction of two velocity vectors, $\sigma_v, \sigma_d$ are normalization factors for velocity and direction such that $\frac{\|\vec{v}_t-\vec{v}_g\|_t}{\sigma_v} \in [0,1], \frac{\|\vec{v}_t-\vec{v}_g\|_d}{\sigma_d} \in [0,1], w_v \in [0,1], w_d \in [0,1], w_v + w_d = 1.0$ are weight factors for velocity and direction. After the decomposition, we can balance the importance of the velocity direction and the magnitude by adjusting the weights $w_v$ and $w_d$.

To analyze the impact of $w_v$ and $w_d$ on policy training, we conduct training with different values of $w_v$ and $w_d$ and present the corresponding results in Fig. 14. From Fig. 14a, it is evident that the smaller the value of $w_v$, the larger the error in velocity magnitude at the end of episode, indicating a positive correlation between the increase in reward weight $w_v$ and the reduction in error for the relatively simple control of velocity magnitude.

Fig. 14b shows that when $w_d$ is set to 0.5, 0.625, and 0.75, the resulting policies have similar errors in velocity direction, but the larger the value of $w_d$, the faster the error decreases during training, suggesting that a higher $w_d$ is more beneficial for the policy to reduce the error in velocity direction. However, when $w_d$ is set to the highest value of 0.875, the error in velocity direction is the largest. This is because the policy is unable to reduce the error in velocity magnitude to within 10, which prevents the policy from triggering RT and limits its ability to decrease the error in velocity direction.

Fig. 14c and 14d illustrate the trends in cumulative reward and success rate. It can be observed that appropriately increasing $w_d$ is beneficial for enhancing the policy's capability to obtain cumulative reward and to speed up the increase in success rate. However, an excessively large $w_d$ affects the error in velocity magnitude, which is detrimental to the improvement of cumulative reward and further restricts the increase in success rate.

## M STATE AND ACTION SMOOTHNESS OF DEMONSTRATIONS

Two metrics are introduced to measure the smoothness of states and actions. Taking the pitch angle $\theta$ as an example, the formulas for these two metrics are given in the following. And the other variables of state and action are the same. The first metric is inspired by Discrete Fourier Transform (DFT),

and is calculated as

$$s_1^\theta(\tau) = \frac{2}{nf_s} \sum_{i=1}^{n} M_i f_i, \tag{23}$$

where $f_s$ is the sampling frequency, $M_i$ and $f_i$ are the amplitude and frequency of the $i^{th}$ frequency component Mysore et al. (2021). Larger values on this metric indicate the presence of larger high-frequency signal components, thus less smoothness the signal is. The second metric is calculated by accumulating the change of variable over the entire trajectory, and is calculated as

$$s_2^\theta(\tau) = \sum_{i=1}^{|\tau|-1} |\theta_{i+1} - \theta_i|. \tag{24}$$

The larger this value, the greater the oscillation of variable between two consecutive timestamps.

Table 12 shows that $\mathcal{D}_E^0$ has nearly the smallest $s_1$ and $s_2$ values among all demonstration sets. This is because the derivative term of the PID controller explicitly suppresses oscillations in the control variables. When IRPO starts updating $\mathcal{D}_E^0$ with the policy trained by RL, both smoothness metrics $s_1$ and $s_2$ begin to increase. This is because we do not include terms explicitly in the reward function to penalize oscillations, and the inadequately trained RL policy is unable to achieve goals at the fastest speed, resulting in some degree of oscillation. However, there is little difference between $s_1$ and $s_2$ across $\mathcal{D}_E^1$, $\mathcal{D}_E^2$, and $\mathcal{D}_E^3$. This indicates that the policies trained with VVC-Gym's reward function do not exhibit significant differences in the smoothness of states and actions.

## N    SUGGESTIONS FOR TRANSFERRING MODELS TRAINED BY VVC-GYM TO REAL WORLD UAVS

We suggest the followings when transferring models trained by VVC-Gym to real world UAVs:

- VVC-Gym employs an open-source UAV model Simulator (2022), so we need to consider the differences between the UAV model in the environment and real-world UAVs.

- We have not yet incorporated environmental dynamic factors into VVC-Gym, such as changes in wind, air temperature, and humidity. Before deploying the model to real-world UAVs, it is necessary to thoroughly evaluate the robustness of the model.

- It is necessary to consider the computational capabilities of the onboard computer to ensure that it can complete the model's inference within the physical time of an environment step.

## O    SOCIETAL IMPACTS

We open-source VVC-Gym and its accompanying demonstrations. They can be utilized to train control policies for fixed-wing UAVs. However, it is important to note that VVC-Gym's design does not account for specific hardware-related requirements, such as the need to avoid (1) bang-bang control to minimize hardware wear and tear, and (2) prolonged overload beyond a certain threshold to ensure the structural integrity of the aircraft within acceptable limits.

