# OpenReview forum: "VVC-Gym: A Fixed-Wing UAV Reinforcement Learning Environment for Multi-Goal Long-Horizon Problems"
_ICLR.cc/2025/Conference — ICLR 2025 Poster_

### Official Review · Reviewer_soQJ · 2024-10-22

**Soundness:** 3
**Presentation:** 3
**Contribution:** 3
**Rating:** 6
**Confidence:** 3

**Summary:**

This paper presents VVC-Gym, a fix-wing UAV RL environment for multi-goal long-horizon tasks. The environment is built upon real dynamics and rooted in a real-world UAV control problem. Various simulation shows VVC-Gym is suitable for studying the influence of environment designs on multi-goal long-horizon problems, the necessity of expert demonstrations on the task and suitable RL algorithm factors for the environment.

**Strengths:**

Fix-wing UAV control inherently requires multi-goal and long-horizon RL. The biggest strength to me is that the problem and the environment is based on real need, not simulations. I also appreciate the explorations of the authors take on how to design the environment and how to make RL work in this context.

**Weaknesses:**

1. I am not a researcher on fixed-wing UAV control and multi-goal long-horizon problem. The biggest problem for me is that while the authors present an environment for future researchers, they failed to give a clear guidance on how to use this environment for benchmark. For example, as a user of the environment, I want to know:

* What are the number of tasks in this environment, and what are their difficulties, respectively? Seems to me there is only one task. In this way, users might not use this environment in their study since it is likely to be rejected by other reviewers by the reason like "small amount of experiments".

* What are the recommended configuration, hyperparameters and algorithms for me to start with, for researchers in the field of (1) GCRL (2) demonstration-based RL and (3) fixed-wing UAV control?

* As for demonstrations, if I want to generate trajectories to support my own task, do I have tools for that in the environment? Are there standardized demonstrations recommended by the authors?

2. If conventional algorithms such as PID/MPC can already solve the task, why should we use RL for this problem? Maybe we can significantly reduce the complexity of the problem by planning?

3. After searching on google, I find some repo doing similar tasks like this: https://github.com/liuqh16/LAG, which is also based on fixed wings. Can you also discuss existing simulators of fixed-wing UAV control?

4. Some minors: in page 7, ... with the results presented in Fig. 2b, which should be Table. 2b. Please also check others.

To sum up, as RL researchers, we always need real-world environments for simulations and I appreciate the large amount of work by authors to develop this environment and make it work. However, I'm not sure how to use the environment without pain. Maybe the authors could revise the paper by giving more recommendations to users?

**Questions:**

Please see "Weakness" section. I will read the rebuttal and adjust scores in reviewer-author discussion.

**Details Of Ethics Concerns:**

None.

---

> ### Author Response · Authors · 2024-11-20
>
> We sincerely appreciate your careful review of our paper and valuable feedback. We address your concerns below.
>
> > `W1: I am not a researcher on fixed-wing UAV control and multi-goal long-horizon problem. The biggest problem for me is that while the authors present an environment for future researchers, they failed to give a clear guidance on how to use this environment for benchmark.`
>
> Thank you for your suggestion. We have also considered how to facilitate researchers' use of our environment and dataset before submitting our manuscript. Therefore, we have included three parts in the [Supplementary Material](https://openreview.net/attachment?id=5xSRg3eYZz&name=supplementary_material) submitted:
>
> 1. **The source code repository of VVC-Gym**, in which we have detailed the installation, configuration, and several usage examples of VVC-Gym in the README.md.
> 2. **The case repository for using VVC-Gym**, which includes scripts for generating demonstrations and training scripts for the baselines mentioned in the manuscript. We have detailed the methods for running these scripts in the README.md.
> 3. **Some visualized demonstrations** of the RL policy we have trained.
>
> Due to the double-blind review mechanism of ICLR, we have not open-sourced our work, including the environment source code, dataset, case source code, and trained policies, etc. Thank you again for your suggestion. We will:
>
> 1. enrich our case repository by providing more baseline algorithms and more detailed usage instructions.
> 2. open-source the environment source code, dataset, case source code, and trained policies, etc.
> 3. add a section in the Appendix of the manuscript to introduce how to conduct research on the proposed environment, dataset, and baselines.
>
> > `W1.1: What are the number of tasks in this environment, and what are their difficulties, respectively? Seems to me there is only one task. In this way, users might not use this environment in their study since it is likely to be rejected by other reviewers by the reason like "small amount of experiments".`
>
> In our submitted manuscript, we introduced the task of Fixed-wing UAV's Velocity Vector Control (VVC). Currently, we have expanded our code to include Attitude Control and several Basic Flight Maneuvers (BFMs), such as level turn and barrel roll. All these tasks belong to multi-goal problems. The extended code can be found at [(link)](https://github.com/ForAnonymousUse/ForICLR2025/blob/main/VVC-Gym.zip). These new tasks are located in the directory VVC-Gym/vvcgym/tasks/.
>
> Taking the VVC task as an example, this task requires the fixed-wing UAV to achieve any arbitrary velocity vector in three-dimensional space. Therefore, in different episodes, the RL agent faces different desired velocity vectors. If the initial velocity vector $s_0$ is fixed, different desired velocity vectors $g$ have different levels of difficulty. We define the difficulty of desired velocity vector based on the offset $d(s_0, g)$ between $s_0$ and $g$. We discretize the entire desired goal space and visualize the difficulty of different desired velocity vectors, the results of which can be seen at [(link)](https://github.com/ForAnonymousUse/ForICLR2025/blob/main/visualization%20of%20goal%20difficulty.pdf).
>
> In summary, we focus on multi-goal problems and aim to train a goal-conditioned policy that can achieve all desired goals. Although VVC is a single task, within this task, there are multiple desired goals with varying levels of difficulty.
>
> > `W1.2: What are the recommended configuration, hyperparameters and algorithms for me to start with, for researchers in the field of (1) GCRL (2) demonstration-based RL and (3) fixed-wing UAV control?`
>
> All baselines mentioned in our manuscript can be found in our case repository in the [Supplementary Material](https://openreview.net/attachment?id=5xSRg3eYZz&name=supplementary_material). Additionally, we provide the default parameters for these algorithms. For GCRL researchers, we recommend starting with SAC, HER, and MEGA; for demonstration-based RL, we suggest starting with GCBC; and for fixed-wing UAV control researchers, we recommend beginning with GCBC+PPO+MEGA.
>
> > `W1.3: As for demonstrations, if I want to generate trajectories to support my own task, do I have tools for that in the environment? Are there standardized demonstrations recommended by the authors?`
>
> In the case repository within the [Supplementary Material](https://openreview.net/attachment?id=5xSRg3eYZz&name=supplementary_material), we provide and describe the methods for generating the 8 demonstration datasets mentioned in the manuscript (please refer to the README.md file in this code repository). We will also open-source these 8 demonstration datasets in the future.

---

> ### Author Response · Authors · 2024-11-20
>
> > `W2: If conventional algorithms such as PID/MPC can already solve the task, why should we use RL for this problem? Maybe we can significantly reduce the complexity of the problem by planning?`
>
> We utilize PID as an example to compare classical control methods with RL. We believe that in the complex nonlinear problem of fixed-wing UAV's velocity vector control, PID struggles to provide high-quality policies. PID is suitable for relatively simple linear systems and models with known dynamics, whereas RL is suitable for complex, dynamic, nonlinear systems, especially when models are unknown. **Therefore, we consider PID and RL to have different areas of application.** **The control of fixed-wing UAV is a typical nonlinear problem [1], and PID struggles to provide high-quality solutions.** In our environment, the PID controller (detailed in Appendix C) achieves only a 20.08% success rate, while the best RL policy achieves a 71.68% success rate. Numerous studies in the field of fixed-wing UAV have also found that RL can yield better policies than PID [1,2,3].
>
> In our research, we find that although the success rate of PID is not as good as that of the RL policy, **the data sampled by the PID controller can serve as demonstrations to assist RL in training**. The following table (results from Table 2 and Table 3 of the manuscript) compares the success rates of PID with RL policies trained under different conditions.
>
> ||PID|RL w/o Pre-train|Pre-train w/ BC|Fine-tune w/ RL|
> |:-:|:-:|:-:|:-:|:-:|
> |Success Rate (%)|20.08|0.04±0.03|17.08±0.57|**38.31±1.62**|
>
> It can be observed that:
>
> * The PID controller has a relatively low success rate.
> * If trained from scratch, the RL policy can barely achieve any goals within a limited training budget.
> * Although the success rate of the policy pre-trained on demonstrations is lower than that of PID, when the policy is fine-tuned with RL, its success rate far exceeds that of PID.
>
> In summary, in the complex nonlinear problem of fixed-wing UAV's velocity vector control, PID, due to its limitations, struggles to achieve goals with high quality, but the data sampled by PID can be used as demonstrations to assist RL in training well-performing policies.
>
> [1] Bøhn E, Coates E M, Reinhardt D, et al. Data-efficient deep reinforcement learning for attitude control of fixed-wing UAVs: Field experiments[J]. IEEE Transactions on Neural Networks and Learning Systems, 2023, 35(3): 3168-3180.
>
> [2] Koch III W F. Flight controller synthesis via deep reinforcement learning[D]. Boston University, 2019.
>
> [3] Eckstein F, Schiffmann W. Learning to fly–building an autopilot system based on neural networks and reinforcement learning[D]. Master's thesis. FernUniversität Hagen, Hagen, Germany, 2020.

---

> ### Author Response · Authors · 2024-11-20
>
> > `W3: After searching on google, I find some repo doing similar tasks like this: https://github.com/liuqh16/LAG, which is also based on fixed wings. Can you also discuss existing simulators of fixed-wing UAV control?`
>
> Existing fixed-wing simulators that support RL training include: Fixed-Wing-Gym [1], Gym-JSBSim [2], Markov-Pilot [3], LAG [4], etc. We discuss the differences between VVC-Gym and these simulators from two perspectives: (1) the simulator itself, and (2) support for RL research:
>
> 1. **From the perspective of the simulator itself**: Our goal is to provide RL researchers with a more extensible and computationally efficient simulator. This allows RL researchers to customize VVC-Gym to their tasks of interest and efficiently validate their algorithm designs.
>
>     (1) **Scalability**. We have made the scalability of VVC-Gym the most important design objective from the beginning. Researchers can easily extend VVC-Gym to investigate new control tasks, new fixed-wing UAV models, and generate new demonstrations, among other applications.
>
>     * *Using other aircraft models*: VVC-Gym uses an open-source, more realistic fixed-wing aircraft model, and can replace the aircraft model with other open-source aircraft models [(link)](https://mirrors.ibiblio.org/flightgear/ftp/Aircraft-2020/) according to actual needs.
>     * *Defining new tasks*: Task and Simulator are decoupled (overall architecture as shown in Fig.1), so when solving new types of tasks, such as Attitude Control, Basic Flight Maneuvers, etc., only new tasks need to be added without modifying existing source code. We have provided some of these new tasks, which can be found at [(link)](https://github.com/ForAnonymousUse/ForICLR2025/blob/main/VVC-Gym.zip) (locate at directory VVC-Gym/vvcgym/tasks/).
>
>     (2) **Efficiency**. For RL researchers, the computational efficiency of the simulator is crucial as it affects the efficiency of validating algorithm designs. Since the core of fixed-wing UAV simulation lies in the computation of the Equations of Motion (EoM), VVC-Gym employs C++ for the EoM calculations, which is more computationally efficient. To demonstrate the computational efficiency of VVC-Gym, we evaluate the FPS of VVC-Gym and Fixed-Wing-Gym on the machine described in Appendix A, and the results are as follows:
>
>     ||Fixed-Wing-Gym|VVC-Gym|
>     |:-:|:-:|:-:|
>     |FPS|1325|2356|
>
>     _**Note**: The above training used the PPO algorithm from the StableBaselines framework, with 64 rollout workers, and $10^6$ environmental steps._
>
>     It can be seen that VVC-Gym has a much higher computational efficiency than the Python-based Fixed-Wing-Gym. Additionally, in Appendix A, we also compared the FPS of VVC-Gym with commonly used RL environments, and the results show that VVC-Gym achieves the sampling speed of environments commonly used in academic research and even surpasses several of them.
>
> 2. **From the perspective of supporting RL algorithm research**:
>
>     ||VVC-Gym|Fixed-Wing-Gym|Gym-JSBSim|Markov-Pilot|LAG|
>     |:-:|:-:|:-:|:-:|:-:|:-:|
>     |Task|VVC, AC, BFMs|AC|Level turn|Gliding descent|AC, 1v1, 2v2|
>     |MDP type|Goal-Augmented MDP, standard MDP|standard MDP|standard MDP|standard MDP|standard MDP, Multi-agent MDP|
>     |Demonstrations|√|☓|☓|☓|☓|
>     |RL baselines|PPO, SAC, HER, GCBC, GCBC+PPO, MEGA, RIG, DISCERN|PPO|PPO|DDPG|PPO, MPPO|
>
>     _**Note**: VVC is the abbreviation for Velocity Vector Control, AC is the abbreviation for Attitude Control, and BFMs is the abbreviation for Basic Flight Maneuvers._
>
>     It can be seen that: Firstly, in terms of tasks, VVC-Gym is the first publicly available RL environment for velocity vector control. Secondly, only VVC-Gym is modeled as a Goal-Augmented MDP, which better supports GCRL research. Thirdly, only VVC-Gym is accompanied by demonstrations, which can support research on demonstration-based RL. Fourthly, VVC-Gym provides baselines for 8 GCRL algorithms, while other environments only provide baselines for 1 to 2 RL algorithms. In summary, VVC-Gym provides stronger support for conducting GCRL research.
>
> [1] Bøhn E, Coates E M, Moe S, et al. Deep reinforcement learning attitude control of fixed-wing uavs using proximal policy optimization[C]//2019 international conference on unmanned aircraft systems (ICUAS). IEEE, 2019: 523-533.
>
> [2] Rennie G. Autonomous control of simulated fixed wing aircraft using deep reinforcement learning[J]. 2018.
>
> [3] Eckstein F, Schiffmann W. Learning to fly–building an autopilot system based on neural networks and reinforcement learning[D]. Master's thesis. FernUniversität Hagen, Hagen, Germany, 2020.
>
> [4] https://github.com/liuqh16/LAG

---

> > ### Comment · Reviewer_soQJ · 2024-11-22
> > **Thanks for the detailed response**
> >
> > Thanks for the detailed response of the author. While I'm not a researcher in this field, the response of authors have solved my concerns to some extent, yet I still wonder we could have some (10~20) standardized settings in final version, if possible. I would recommend authors to provide a clear guidance for users on how to use the repo in the final version. I have raised my score.

---

> > > ### Author Response · Authors · 2024-11-23
> > >
> > > Dear Reviewer soQJ,
> > >
> > > We sincerely appreciate the time and effort you invested in reviewing our manuscript. Your insightful comments and concerns have greatly contributed to the improvement of our paper. We are sure to provide a clear guidance for users on how to use the proposed environment and demonstrations in the final version.
> > >
> > > Thank you once again for your valuable feedback.

---

### Official Review · Reviewer_ecAp · 2024-10-24

**Soundness:** 2
**Presentation:** 2
**Contribution:** 2
**Rating:** 5
**Confidence:** 4

**Summary:**

This paper propose a multi-goal long-horizon Reinforcement Learning (RL) environment based on realistic fixed-wing UAV’s velocity vector control, named VVC-Gym, and generate multiple demonstration sets of various quality. I suggest that the authors improve their academic writing skills, especially in organizing their ideas properly and expressing their points of view.

**Strengths:**

This paper propose a multi-goal long-horizon Reinforcement Learning (RL) environment based on realistic fixed-wing UAV’s velocity vector control, named VVC-Gym, and generate multiple demonstration sets of various quality.

**Weaknesses:**

I suggest that the authors improve their academic writing skills, especially in organizing their ideas properly and expressing their points of view. For example, the paper first presents the concept of multi-goal long-horizon problems. It does not clearly define the problem or list related works to support the issues, but it spends many words discussing the GCRL, which confuses me about the paper's writing logic. If the authors think the multi-goal long-horizon problems belong to the GCRL, they need to discuss the background of the GCRL and the current state-of-the-art. Then, they need to analyze their relationships and provide recent research on the challenge. Moreover, I believe this problem might belong to the multi-objective multi-agent area. Please check the paper:
Rădulescu, R., Mannion, P., Roijers, D.M. et al. Multi-objective multi-agent decision making: a utility-based analysis and survey. Auton Agent Multi-Agent Syst 34, 10 (2020).

I feel it is more like a technical paper introducing a platform in reinforcement learning than a research paper. Additionally, I suggest the authors add more baseline algorithms to their experiments, not just PPO.

**Questions:**

Generally, I strongly recommend several papers, as shown below, in which authors can learn how to improve academic writing skills and organize corresponding ideas from them.

1) Yang, Q., & Parasuraman, R. Bayesian strategy networks based soft actor-critic learning. ACM Transactions on Intelligent Systems and Technology (TIST).

2) Mannion P, Devlin S, Duggan J, Howley E. Reward shaping for knowledge-based multi-objective multi-agent reinforcement learning. The Knowledge Engineering Review. 2018

---

> ### Author Response · Authors · 2024-11-21
>
> Thank you for your valuable feedback on our work. We appreciate the opportunity to address your concerns.
>
> > `Q1: Generally, I strongly recommend several papers, as shown below, in which authors can learn how to improve academic writing skills and organize corresponding ideas from them.`
>
> Thank you for your suggestions on our manuscript. Our current writing thought is structured as follows: we begin by introducing the concept of multi-goal long-horizon problems, followed by an exploration of the challenges that arise when attempting to solve these problems using Goal-Conditioned Reinforcement Learning (GCRL). We then explain why current multi-goal environments fall short in supporting GCRL researchers in tackling these challenges. Subsequently, we present VVC-Gym in detail, outlining its MDP definition and demonstrations. Finally, we illustrate how VVC-Gym can facilitate various research endeavors for GCRL researchers through the introduction of baselines and the conduct of ablation studies.
>
> Additionally, in the appendix, we provide a detailed introduction to Goal-Augmented MDP (the mathematical definition of the problems solved by GCRL), the calculation of aircraft aerodynamic equations, the generation method of demonstrations, and more ablation studies, etc.
>
> Thank you for your suggestions again. We will add one additional page to the manuscript and adjust the sequence of some content in the main text and Appendix to ensure the coherence of the paper.

---

> ### Author Response · Authors · 2024-11-21
>
> > `W1: Moreover, I believe this problem might belong to the multi-objective multi-agent area. Please check the paper: Rădulescu, R., Mannion, P., Roijers, D.M. et al. Multi-objective multi-agent decision making: a utility-based analysis and survey. Auton Agent Multi-Agent Syst 34, 10 (2020).`
>
> **We focus on single-agent multi-goal problems:**
>
> 1. **Our manuscripts focuses on "multi-goal RL"**. There is a fundamental difference between "multi-goal RL" and "multi-objective RL":
>
> * **Multi-Objective RL (MORL)**: In MORL, the reward is a vector [1]. In other words, the agent pursues multiple objectives within a single episode [1]. A simple example is that the agent must maximize a reward $r_1 \in [0, 1]$ indicating the completion of the task and minimize a penalty $r_2 \in [-1,0]$ based on control gain or energy consumption [2]. The reward is a vector $\mathbf{r} = [r_1,r_2]$. Since RL algorithms rely on scalar rewards, a preference function $f$ is needed to convert the vector reward into a scalar for optimization. In the example, a linear preference function $f(\mathbf{r}) = 0.8 r_1 + 0.2 r_2$ can be used to transform the reward vector into a scalar reward. In MORL, the policy is trained based on the reward vector and the preference function. When the preference is known, the problem reduces to standard RL. When the preference is unknown, the main task of MORL is to find Pareto-optimal solutions [1].
>
> * **Multi-goal RL** (also known as **goal-conditioned RL** or **goal-oriented RL**): The reward remains scalar [3]. In multi-goal RL, the agent pursues a single objective within an episode, which is to achieve a specific desired goal. The concept of "multi-goal" is reflected in that the agent may be required to achieve different desired goals across different episodes. The objective of "multi-goal RL" is to obtain a policy that is capable of achieving any desired goal. For example, a robotic arm may be tasked with reaching a point 1 meter to the left (goal $g_1$) in one episode and a point 0.5 meters to the right (goal $g_2$) in another episode. We aim for an agent capable of achieving any desired goal ($\pi$ that can achieve both $g_1$ and $g_2$), rather than using different agents for different goals ($\pi_1$ for $g_1$ and $\pi_2$ for $g_2$). From the reward type perspective, multi-goal RL is consistent with standard RL, both being scalar. The only difference is that multi-goal RL uses a goal-conditioned reward $r_g : \mathcal{S} \times \mathcal{A} \rightarrow \mathbb{R}, g \in \mathcal{G} $. The objective of multi-goal RL becomes optimizing the standard RL objective over the desired goal distribution, $ E_{g \sim p_{dg}, \pi}[ \sum_{t=0}^{\infty} \gamma^t r_g (s_t) ] $.
>
> * **Velocity Vector Control**: The objective is to enable the fixed-wing UAV to achieve any desired goal velocity vector. However, at any given moment, the fixed-wing UAV has only one goal velocity vector to pursue. Therefore, Velocity Vector Control falls under the category of typical multi-goal problems.
>
> **In summary, multi-objective RL and multi-goal RL represent different RL research fields. Velocity Vector Control is a multi-goal problem, and Our focus is on multi-goal RL.**
>
> 2. Our manuscript does not include contents related to multi-agent settings. From the application perspective, **we focus on solving the control problem of _a single fixed-wing UAV_**.
>
> [1] Yang R, Sun X, Narasimhan K. A generalized algorithm for multi-objective reinforcement learning and policy adaptation[J]. Advances in neural information processing systems, 2019, 32.
>
> [2] Bøhn E, Coates E M, Reinhardt D, et al. Data-efficient deep reinforcement learning for attitude control of fixed-wing UAVs: Field experiments[J]. IEEE Transactions on Neural Networks and Learning Systems, 2023, 35(3): 3168-3180.
>
> [3] Liu M, Zhu M, Zhang W. Goal-conditioned reinforcement learning: Problems and solutions[J]. IJCAI International Joint Conference on Artificial Intelligence, 2022.

---

> ### Author Response · Authors · 2024-11-21
>
> > `W2: I feel it is more like a technical paper introducing a platform in reinforcement learning than a research paper.`
>
> Our work falls under the category of datasets and benchmarks. We are dedicated to providing the GCRL community with: (1) **an environment** for studying multi-goal long-horizon problems; (2) **demonstrations** to assist in GCRL training; and (3) **baselines** on some GCRL algorithms. Our core contribution is to help future GCRL researchers investigate how to better solve multi-goal long-horizon problems. We collect and study in detail some related work on environments, datasets, and benchmarks [1,2,3,4,5] from **ICLR 2024**, and we believe that **our work meets the requirements for research papers in the Datasets and Benchmark Track**.
>
> [1] Bonnet C, Luo D, Byrne D J, et al. Jumanji: a Diverse Suite of Scalable Reinforcement Learning Environments in JAX[C]//The Twelfth International Conference on Learning Representations, 2024.
>
> [2] Tec M, Trisovic A, Audirac M, et al. SpaCE: The Spatial Confounding Environment[C]//The Twelfth International Conference on Learning Representations, 2024.
>
> [3] Huang S, Weng J, Charakorn R, et al. Cleanba: A Reproducible and Efficient Distributed Reinforcement Learning Platform[C]//The Twelfth International Conference on Learning Representations. 2024.
>
> [4] Zhou S, Xu F F, Zhu H, et al. Webarena: A realistic web environment for building autonomous agents[C]//The Twelfth International Conference on Learning Representations. 2024.
>
> [5] Yuan Y, Hao J, Ma Y, et al. Uni-RLHF: Universal Platform and Benchmark Suite for Reinforcement Learning with Diverse Human Feedback[C]//The Twelfth International Conference on Learning Representations. 2024.
>
> > `W3: Additionally, I suggest the authors add more baseline algorithms to their experiments, not just PPO.`
>
> Thank you for your suggestion. In our manuscript, Section 4.3 reports the results of a total of **8 algorithms**, including PPO, SAC, HER, GCBC, GCBC+PPO, MEGA, RIG, and DISCERN. The corresponding training scripts are included in the [Supplementary Material](https://openreview.net/attachment?id=5xSRg3eYZz&name=supplementary_material) we submitted. Additionally, the table below shows the baselines provided by other Fixed-wing UAV environments, indicating that we include a larger number of baselines compared to other environments. We are currently evaluating Goal-conditioned offline RL [1] algorithms on VVC-Gym, and we will extend our code repository to include the relevant training scripts and evaluating results in the future.
>
> ||VVC-Gym|Fixed-Wing-Gym [2]|Gym-JSBSim [3]|Markov-Pilot [4]|LAG [5]|
> |:-:|:-:|:-:|:-:|:-:|:-:|
> |RL baselines|PPO, SAC, HER, GCBC, GCBC+PPO, MEGA, RIG, DISCERN|PPO|PPO|DDPG|PPO, MPPO|
>
> [1] Park S, Frans K, Eysenbach B, et al. OGBench: Benchmarking Offline Goal-Conditioned RL[J]. arxiv preprint arxiv:2410.20092, 2024.
>
> [2] Bøhn E, Coates E M, Moe S, et al. Deep reinforcement learning attitude control of fixed-wing uavs using proximal policy optimization[C]//2019 international conference on unmanned aircraft systems (ICUAS). IEEE, 2019: 523-533.
>
> [3] Rennie G. Autonomous control of simulated fixed wing aircraft using deep reinforcement learning[J]. 2018.
>
> [4] Eckstein F, Schiffmann W. Learning to fly–building an autopilot system based on neural networks and reinforcement learning[D]. Master's thesis. FernUniversität Hagen, Hagen, Germany, 2020.
>
> [5] https://github.com/liuqh16/LAG

---

> ### Author Response · Authors · 2024-11-24
>
> Dear Reviewer ecAp:
>
> We are approaching the conclusion of the author-reviewer discussion period. Should you have any further questions or require clarification on any points, please feel free to reach out. We are committed to addressing your queries promptly. We greatly value your feedback and look forward to your insights.

---

> > ### Comment · Area_Chair_CCAi · 2024-11-24
> > **Please respond to rebuttal ASAP**
> >
> > Dear reviewer,
> > The process only works if we engage in discussion. Can you please respond to the rebuttal provided by the authors ASAP?

---

> ### Author Response · Authors · 2024-12-01
>
> Dear Reviewer ecAP,
>
> We wish to remind you that the author-reviewer discussion period concludes in only one day. We kindly request that you review the rebuttal. Should you have further questions or concerns, please rest assured that we are committed to addressing them promptly. Your feedback is immensely valuable to us, and we eagerly await your insights.

---

### Official Review · Reviewer_exNf · 2024-11-04

**Soundness:** 3
**Presentation:** 3
**Contribution:** 3
**Rating:** 8
**Confidence:** 3

**Summary:**

This paper proposes a multi-goal long-horizon Reinforcement Learning (RL) environment based on realistic fixed-wing UAV velocity vector control, named VVC-Gym. The proposed environment is studied through various ablation studies using different Goal-Conditioned RL (GCRL) algorithms and is equipped with multi-quality demonstration sets. Baselines are also provided on the environment.

**Strengths:**

1. The proposed benchmark is useful for the GCRL community.

2. The benchmark is equipped with demonstrations with different quality and several baseline algorithms.

3. The influence of the environment-related parameters is studied via ablations.

4. Baseline methods are provided with the benchmark.

5. The paper is well-written and easy to read.

**Weaknesses:**

1. Some writing issues:

    - The citation format is not very suitable. The authors may consider using `\citep` for most of the citations.

    - Line 349 and Line 357: I guess Fig. 2a and Fig. 2b should be Table. 2a and Table. 2b.

    - Figure 5 is provided but not mentioned in the text.

2. The only task provided is the velocity vector control. If various types of different tasks are provided, the paper will have a larger influence and contribution.

**Questions:**

1. In Fig. 6c, we can conclude that there are several distinct stages in training where different termination conditions are triggered. However, what can we conclude from Fig. 6d?

2. Seems that the success rates of all the baseline methods are often low (less than 50%). What might be the possible reasons that the GCRL algorithms fail to achieve a higher success rate?

---

> ### Author Response · Authors · 2024-11-20
>
> Thank you for your thorough review and valuable feedback on our work. We have addressed the three issues you raised and address your concerns in the following:
>
> > `Q1: In Fig. 6c, we can conclude that there are several distinct stages in training where different termination conditions are triggered. However, what can we conclude from Fig. 6d?`
>
> We aim to explain why the application of termination conditions can enhance exploration efficiency through Fig.6d. Taking **ES** (Extreme State, where the episode is terminated when an unreasonable extreme state occurs, such as excessive roll angular rate) and **T** (Timeout, where the episode is terminated after reaching the maximum simulation length of max_episode_length steps) as examples, we provide the following explanation:
>
> * Without ES, all failed episodes would be simulated up to the predefined maximum simulation length of 400 steps in T. Even if unreasonable extreme states occur, the environment would still collect subsequent meaningless transitions until reaching 400 steps.
> * With ES, during training, ES terminates the corresponding unreasonable episodes around 200-250 steps. This means that for each trajectory with an extreme state, employing ES can save 150-200 steps. Therefore, the application of ES can, to some extent, avoid collecting meaningless transitions like extreme states, thus improving exploration efficiency.

---

> ### Author Response · Authors · 2024-11-20
>
> > `Q2: Seems that the success rates of all the baseline methods are often low (less than 50%). What might be the possible reasons that the GCRL algorithms fail to achieve a higher success rate?`
>
> We address your concern from the following three aspects:
>
> 1. **Fixed-wing UAV's velocity vector control (VVC) is a challenging task.** The difficulties lie in:
>
> * The large exploration space of the policy, which is a continuous state, continuous action problem, and the policy requires additional exploration of the goal space during training.
> * The long interaction sequences, with the average length of demonstrations exceeding 280. Even well-trained policies require an average of over 100 steps to achieve a goal, and more challenging goals can demand upwards of 300 steps (see Table 1 in the manuscript for corresponding experimental data).
>
>     We provide evidence that VVC is a challenging task through the following three sets of experiments:
>
>     (1) ***Standard RL algorithms struggle to solve the VVC task.*** The table below shows the % success rates of SAC and PPO. It is evident that SAC and PPO struggle to solve the VVC task.
>
>     ||SAC|PPO|
>     |:-:|:-:|:-:|
>     |VVC|1.08±0.48|0.04±0.03|
>
>     (2) ***Existing GCRL algorithms can effectively solve common multi-goal tasks in academic research, but they can only solve the VVC task to a certain extent.*** We compare the performance of different GCRL algorithms on VVC and common multi-goal tasks in academic research, PointMaze (PointMaze_Large_DIVERSE_G-v3 [(link)](https://robotics.farama.org/envs/maze/point_maze/)) and Reach (PandaReach-v3 [(link)](https://panda-gym.readthedocs.io/en/latest/usage/environments.html)). The results are shown in the table below. It can be seen that these GCRL algorithms can almost completely solve Reach and PointMaze, but the best algorithm achieves only a 38.31% success rate on VVC. These results indirectly reflect that VVC is a challenging task.
>
>     ||MEGA|GCBC|GCBC+PPO|
>     |:-:|:-:|:-:|:-:|
>     |Reach|100.0±0.0|70.63±2.99|100.0±0.0|
>     |PointMaze|100.0±0.0|75.96±5.34|93.33±3.06|
>     |VVC|8.32±1.86|17.08±0.57|38.31±1.62|
>
>     **Note**: *The demonstrations used in the Reach experiments are from the official script provided by Panda-Gym [(link)](https://panda-gym.readthedocs.io/en/latest/usage/manual_control.html), the demonstrations used in the PointMaze experiments are from Minari [(link)](https://minari.farama.org/datasets/D4RL/pointmaze/), and the demonstrations used in the VVC experiments are $\mathcal{D}_E^0$ from the manuscript.*
>
>     (3) ***The human-designed classical PID controller (detailed in Appendix C of the manuscript) has only a 20.08% success rate***, which also indirectly reflects that VVC is a challenging task.
>
> 2. **Although the success rates of these GCRL algorithms are less than 50%, the difference between using and not using GCRL is significant.** The table below shows some comparison results (data from Table 2 in the manuscript). Taking MEGA as an example, when using MEGA to sample behavioral goals during training, the algorithm's success rate can be increased from 38.31% to 48.62%, achieving a 26.91% improvement.
>
>     |Algorithm|GCRL methods|Success Rate|
>     |:-:|:-:|:-:|
>     |SAC|w/o|1.08±0.48|
>     |SAC|w/ HER|8.32±1.86|
>     |PPO|w/o|0.04±0.03|
>     |PPO|w/ GCBC|38.31±1.62|
>     |PPO|w/ GCBC+MEGA|48.62±2.35|
>
> 3. We believe that for academic research, the difficulty of the task should progress in tandem with the research on algorithms. **The task should have appropriate levels of difficulty to properly evaluate different algorithms** [1]. If the task is too easy, too hard, or unsolvable, it will fail to provide a useful signal for benchmarking [1]. Therefore, we believe that the current success rates of GCRL algorithms on VVC being less than 50% is helpful for researchers to discover more insights when designing algorithms.
>
> [1] Park S, Frans K, Eysenbach B, et al. OGBench: Benchmarking Offline Goal-Conditioned RL[J]. arxiv preprint arxiv:2410.20092, 2024.

---

> ### Author Response · Authors · 2024-11-20
>
> > `W: The only task provided is the velocity vector control. If various types of different tasks are provided, the paper will have a larger influence and contribution.`
>
> Thank you for your suggestion. In this paper, our main goal is to provide the GCRL community with a task of appropriate difficulty level, along with demonstrations and baselines. Therefore, we choose the velocity vector control task. As you suggested, providing various types of different tasks can indeed make more contributions to the RL community. **We have extended VVC-Gym based on existing interfaces, providing attitude control and several Basic Flight Maneuvers (BFM) tasks**, including level turn and barrel roll. The extended code can be found at [(link)](https://github.com/ForAnonymousUse/ForICLR2025/blob/main/VVC-Gym.zip). These new tasks locate at directory *VVC-Gym/vvcgym/tasks/*. Additionally, we are working to provide more BFM tasks, including Half Cuban Eight, Immelmann, and others.

---

> > ### Comment · Reviewer_exNf · 2024-11-22
> >
> > I would like to thank the authors for the response. Have you updated the paper to include the additional results and to address the writing issues I proposed?

---

> ### Author Response · Authors · 2024-11-23
>
> Thank you for your valuable feedback and prompt response. We have added clarifications for Fig. 6(d) in lines 418-427, corrected the three writing issues, and introduced a new section (Appendix J) that discusses the challenges of the VVC task and presents relevant experimental results. We have re-uploaded the manuscript. We will continue to proofread our manuscript and ensure to introduce the new tasks discussed above in the final version. We are grateful once again for your careful review of our manuscript.

---

> > ### Comment · Reviewer_exNf · 2024-11-25
> >
> > Thanks for uploading the revised manuscript. I'll keep my positive score.

---

> > > ### Author Response · Authors · 2024-11-25
> > >
> > > Dear Reviewer exNf,
> > >
> > > We express our heartfelt gratitude for the time and dedication you've invested in reviewing our manuscript. Your constructive feedback and perceptive comments have been immensely valuable in refining our paper.
> > >
> > > Thank you once again for your valuable feedback.

---

### Official Review · Reviewer_HNY5 · 2024-11-04

**Soundness:** 3
**Presentation:** 3
**Contribution:** 3
**Rating:** 8
**Confidence:** 3

**Summary:**

The Paper presents a Multi-Goal Long Horizon RL Environment based on Fixed Wing UAV Velocity Vector control. The paper further provides a set of various demonstrations, analyzing the quantity and quality of the demonstration and their effect of training. The paper claims that the environment presented is suitable for studing curriculum learning that combined imitation learning and RL, influence of the environment designs on multi-goal long-horizon problems.

**Strengths:**

The paper provides a detailed analysis of the presented environment, including evaluation of Demonstrations and training of various RL Algorithms with and without Curriculum Methods.

Clear problem formulation and transition function.

Detailed Appendix and Supplementary Materials containing demonstrations.

**Weaknesses:**

- It is unclear what the novelty of the environment is compared to other simulation based fixed wing control environments.

- It is unclear if the machine generated demonstrations can be substituted by human demonstrations.

**Questions:**

- It is unclear how the environment would work with the presence of Adversarial Fixed Wing Agents [1, 2].

- Is the environment applicable to Inverse Reinforcement Learning applications?

- How much reward engineering is required to train an agent to learn a different policy than what is provided?

- What are the challenges of transfering models trainined in this environment to real world UAVs?

[1] Strickland L. G., “Coordinating Team Tactics for Swarm-vs-Swarm Adversarial Games,” Ph.D. Thesis, Georgia Inst. of Technology, Atlanta, GA, July 2022, [https://smartech.gatech.edu/handle/1853/67090](https://smartech.gatech.edu/handle/1853/67090)
[2] B. Vlahov, E. Squires, L. Strickland and C. Pippin, "On Developing a UAV Pursuit-Evasion Policy Using Reinforcement Learning," 2018 17th IEEE International Conference on Machine Learning and Applications (ICMLA), Orlando, FL, USA, 2018, pp. 859-864, doi: 10.1109/ICMLA.2018.00138.

---

> ### Author Response · Authors · 2024-11-20
>
> We sincerely appreciate your careful review of our paper and valuable feedback. We address your concerns below.
>
> > `W1: It is unclear what the novelty of the environment is compared to other simulation based fixed wing control environments.`
>
> Existing fixed-wing simulators that support RL training include: Fixed-Wing-Gym [1], Gym-JSBSim [2], Markov-Pilot [3], LAG [4], etc. We discuss the differences between VVC-Gym and these simulators from two perspectives: (1) the simulator itself, and (2) support for RL research:
>
> 1. **From the perspective of the simulator itself**: Our goal is to provide RL researchers with a more extensible and computationally efficient simulator. This allows RL researchers to customize VVC-Gym to their tasks of interest and efficiently validate their algorithm designs.
>
>     (1) **Scalability**. We have made the scalability of VVC-Gym the most important design objective from the beginning. Researchers can easily extend VVC-Gym to investigate new control tasks, new fixed-wing UAV models, and generate new demonstrations, among other applications.
>
>     * *Using other aircraft models*: VVC-Gym uses an open-source, more realistic fixed-wing aircraft model, and can replace the aircraft model with other open-source aircraft models [(link)](https://mirrors.ibiblio.org/flightgear/ftp/Aircraft-2020/) according to actual needs.
>     * *Defining new tasks*: Task and Simulator are decoupled (overall architecture as shown in Fig.1), so when solving new types of tasks, such as Attitude Control, Basic Flight Maneuvers, etc., only new tasks need to be added without modifying existing source code. We have provided some of these new tasks, which can be found at [(link)](https://github.com/ForAnonymousUse/ForICLR2025/blob/main/VVC-Gym.zip) (locate at directory VVC-Gym/vvcgym/tasks/).
>
>     (2) **Efficiency**. For RL researchers, the computational efficiency of the simulator is crucial as it affects the efficiency of validating algorithm designs. Since the core of fixed-wing UAV simulation lies in the computation of the Equations of Motion (EoM), VVC-Gym employs C++ for the EoM calculations, which is more computationally efficient. To demonstrate the computational efficiency of VVC-Gym, we evaluate the FPS of VVC-Gym and Fixed-Wing-Gym on the machine described in Appendix A, and the results are as follows:
>
>     ||Fixed-Wing-Gym|VVC-Gym|
>     |:-:|:-:|:-:|
>     |FPS|1325|2356|
>
>     _**Note**: The above training used the PPO algorithm from the StableBaselines framework, with 64 rollout workers, and $10^6$ environmental steps._
>
>     It can be seen that VVC-Gym has a much higher computational efficiency than the Python-based Fixed-Wing-Gym. Additionally, in Appendix A, we also compared the FPS of VVC-Gym with commonly used RL environments, and the results show that VVC-Gym achieves the sampling speed of environments commonly used in academic research and even surpasses several of them.
>
> 2. **From the perspective of supporting RL algorithm research**:
>
>     ||VVC-Gym|Fixed-Wing-Gym|Gym-JSBSim|Markov-Pilot|LAG|
>     |:-:|:-:|:-:|:-:|:-:|:-:|
>     |Task|VVC, AC, BFMs|AC|Level turn|Gliding descent|AC, 1v1, 2v2|
>     |MDP type|Goal-Augmented MDP, standard MDP|standard MDP|standard MDP|standard MDP|standard MDP, Multi-agent MDP|
>     |Demonstrations|√|☓|☓|☓|☓|
>     |RL baselines|PPO, SAC, HER, GCBC, GCBC+PPO, MEGA, RIG, DISCERN|PPO|PPO|DDPG|PPO, MPPO|
>
>     _**Note**: VVC is the abbreviation for Velocity Vector Control, AC is the abbreviation for Attitude Control, and BFMs is the abbreviation for Basic Flight Maneuvers._
>
>     It can be seen that: Firstly, in terms of tasks, VVC-Gym is the first publicly available RL environment for velocity vector control. Secondly, only VVC-Gym is modeled as a Goal-Augmented MDP, which better supports GCRL research. Thirdly, only VVC-Gym is accompanied by demonstrations, which can support research on demonstration-based RL. Fourthly, VVC-Gym provides baselines for 8 GCRL algorithms, while other environments only provide baselines for 1 to 2 RL algorithms. In summary, VVC-Gym provides stronger support for conducting GCRL research.
>
> [1] Bøhn E, Coates E M, Moe S, et al. Deep reinforcement learning attitude control of fixed-wing uavs using proximal policy optimization[C]//2019 international conference on unmanned aircraft systems (ICUAS). IEEE, 2019: 523-533.
>
> [2] Rennie G. Autonomous control of simulated fixed wing aircraft using deep reinforcement learning[J]. 2018.
>
> [3] Eckstein F, Schiffmann W. Learning to fly–building an autopilot system based on neural networks and reinforcement learning[D]. Master's thesis. FernUniversität Hagen, Hagen, Germany, 2020.
>
> [4] https://github.com/liuqh16/LAG

---

> ### Author Response · Authors · 2024-11-20
>
> > `W2: It is unclear if the machine generated demonstrations can be substituted by human demonstrations.`
>
> **We believe that human demonstrations cannot replace machine-generated demonstrations. Both of them play a crucial role in RL training.** The design of the PID controller encapsulates expert knowledge, and while the collected demonstrations may not be optimal, it offers good interpretability and is suitable for data collection on physical systems. Moreover, it allows for the collection of a large amount of demonstrations. Compared to PID, human expert data may be of higher quality in terms of achieving desired goals but is limited in quantity and may contain some jitter[1]. Therefore, as two manifestations of human expert knowledge, collecting from human and human-designed classical controllers, such as PID, each have their own advantages and disadvantages.
>
> We provide further experiment results on human demonstrations below. We have demonstrations collected from human experts, totaling 613 demonstrations, which corresponds to approximately 2.5 hours of data. [(link)](https://github.com/ForAnonymousUse/ForICLR2025/tree/main/pilot_trajs) provides four screenshots of these human demonstrations. We trained policies with GCBC+PPO on these human play demonstrations and compared their performance with policies trained on demonstrations generated by the PID controller. The results of this comparison are presented in the table below.
>
> |Data Source|Data Collecting Time|Demonstration Quantity $\uparrow$|Average Demonstration Length $\downarrow$|Policy Success Rate $\uparrow$|
> |:-:|:-:|:-:|:-:|:-:|
> |Human|2.5(h)|613|143.71±23.91|0.29±0.02|
> |PID|10.1(min)|10264|281.83±149.48|0.38±0.02|
>
> The results indicate that:
>
> * Human demonstrations have shorter lengths (indicating faster goal achievement), thus they are of higher quality.
> * Collecting 613 human demonstrations took approximately 2.5 hours, whereas collecting 10264 demonstrations with PID took only 10.1 minutes. Therefore, using PID allows for more efficient collection of demonstrations.
> * In terms of policy performance, demonstrations generated by the PID controller can assist RL in achieving higher success rates. On the other hand, although the number of human demonstrations is only $613 / 10264 = 5.97$% of the PID demonstrations, they can assist the RL policy in achieving a success rate that is $0.29 / 0.38 = 76.32$% of the PID.
>
> In summary, in our scenario, human demonstrations are fewer but of higher quality, while machine-generated demonstrations are of lower quality but in greater quantity. The above experiments show that both forms of demonstrations can effectively assist in RL training. Therefore, we believe that human demonstrations cannot replace machine-generated demonstrations. Both of them play a crucial role in RL training.
>
> > `Q1: It is unclear how the environment would work with the presence of Adversarial Fixed Wing Agents [1, 2].`
>
> In this manuscript, we have not considered the topic of adversarial agents in our contributions.
>
> From an algorithmic perspective, we aim to provide Goal-Conditioned RL researchers with an environment, dataset, and baselines that facilitate the study of multi-goal long-horizon problems. From an application perspective, we focus on solving the inner-loop control problem of a single fixed-wing UAV.
>
> Although we have not yet focused on multi-agent adversarial scenarios, our environment supports extension to multi-agent environments. Based on the current architecture (detailed in Fig.1 of our manuscript), a _situation management module_ can be established above the task module to manage multiple agents. We recommend referring to [1] for reward design and training adversarial RL agents.
>
> [1] https://github.com/liuqh16/LAG
>
> > `Q2: Is the environment applicable to Inverse Reinforcement Learning applications?`
>
> Due to the inclusion of demonstrations, VVC-Gym is applicable for studying Inverse Reinforcement Learning (IRL). We provide training scripts for two IRL algorithms on VVC-Gym: modeling reward function with Kernel Density [1], AIRL [2]. The training scripts are available at [(link)](https://github.com/ForAnonymousUse/ForICLR2025/tree/main/eval_on_irl), the training logs are available at [(link)](https://github.com/ForAnonymousUse/ForICLR2025/tree/main/eval_on_irl/logs), and the visualizations of the training results are available at [(link)](https://github.com/ForAnonymousUse/ForICLR2025/tree/main/eval_on_irl/screenshots).
>
> Thank you again for your suggestion. We will continue to expand on these experiments and include the relevant content in our manuscript.
>
> [1] https://imitation.readthedocs.io/en/latest/algorithms/density.html
>
> [2] Fu J, Luo K, Levine S. Learning Robust Rewards with Adverserial Inverse Reinforcement Learning[C]//International Conference on Learning Representations. 2018.

---

> ### Author Response · Authors · 2024-11-20
>
> > `Q3: How much reward engineering is required to train an agent to learn a different policy than what is provided?`
>
> We address your concern from two aspects: 1. The specific workload involved in our reward engineering; 2. The reason behind designing the reward function in a seemingly complex manner.
>
> 1. This is a practical question. We believe that the design of the reward should align with the actual application, so the workload of reward engineering depends on the requirements of the application. Taking the fixed-wing UAV's velocity vector control (VVC) as an example, our reward engineering work mainly covers the content shown in the table below. It can be seen that we conducted a total of $5 \times 1 \times 5= 25$ groups of experiments to find the optimal parameter combinations in the reward function.
>
>     |Parameters in Reward|Physical meaning of parameters|Whether to search|Search set|
>     |:-:|:-:|:-:|:-:|
>     |$w_d, w_v$|Weights of the error of direction and magnitude of the velocity vector|√|(0.25, 0.75), (0.375, 0.625), (0.5, 0.5), (0.625, 0.375), (0.75, 0.25)|
>     |$\delta_d, \delta_v$|Normalization factors for the error of direction and magnitude of the velocity vector|☓|(180.0, 100.0)|
>     |b|Exponential scaling factor for the error of direction and magnitude of the velocity vector|√|0.25, 0.5, 1.0, 2.0, 4.0|
>
>     Among them, the best-performing set is $w_d=0.75, w_v=0.25, b=0.5$. We use this set of parameters as the default parameters for VVC-Gym. We believe that **for GCRL or RL researchers, using the default parameters we provide is sufficient**.
>
> 2. The reward form commonly used in multi-goal environments is $r(s) = -d(s,g)$ [1], where $d(\cdot, \cdot)$ is some distance metric. We found that this reward is very broad and does not work well when directly applied to VVC. The reasons are: (1) The importance of eliminating errors in various parts of the goal is different; (2) The rate of change of the reward function on the error ($b$) needs to be comprehensively designed according to factors such as the length of the control sequence and the precision of judgment of arrival (we have detailed discussions in Section 3.3 of the manuscript). Through experiments, we found that these different settings of the reward will simultaneously affect the training process and the final results. Therefore, we hope that **VVC-Gym can serve as a testbed for RL researchers to explore the impact of different reward settings on training**.
>
> [1] Liu M, Zhu M, Zhang W. Goal-conditioned reinforcement learning: Problems and solutions[J]. IJCAI International Joint Conference on Artificial Intelligence, 2022.
>
> > `Q4: What are the challenges of transfering models trainined in this environment to real world UAVs?`
>
> We list three typical challenges below:
>
> 1. VVC-Gym employs an open-source UAV model [1], so we need to consider the differences between the UAV model in the environment and real-world UAVs.
> 2. We have not yet incorporated environmental dynamic factors into VVC-Gym, such as changes in wind, air temperature, and humidity. Before deploying the model to real-world UAVs, it is necessary to thoroughly evaluate the robustness of the model.
> 3. It is necessary to consider the computational capabilities of the onboard computer to ensure that it can complete the model's inference within the physical time of an environment step.
>
> Thank you again for your suggestions. We will incorporate this discussion into our manuscript to help application-focused researchers understand the challenges of applying VVC-Gym in a production environment.
>
> [1] https://mirrors.ibiblio.org/flightgear/ftp/Aircraft-2020/

---

> ### Author Response · Authors · 2024-11-24
>
> Dear Reviewer HNY5:
>
> We are approaching the conclusion of the author-reviewer discussion period. Should you have any further questions or require clarification on any points, please feel free to reach out. We are committed to addressing your queries promptly. We greatly value your feedback and look forward to your insights.

---

> > ### Comment · Area_Chair_CCAi · 2024-11-24
> > **Please respond to rebuttal ASAP**
> >
> > Dear reviewer,
> > The process only works if we engage in discussion. Can you please respond to the rebuttal provided by the authors ASAP?

---

> ### Comment · Reviewer_HNY5 · 2024-11-26
>
> I would like to thank the authors for their detailed response to the questions regarding applications and novelty. I have updated my scores.

---

> > ### Author Response · Authors · 2024-11-27
> >
> > Dear Reviewer HNY5,
> >
> > We sincerely appreciate the time and effort you invested in reviewing our manuscript. Your insightful comments and concerns have greatly contributed to the improvement of our paper. We are sure to add these discussions to our final version.
> >
> > Thank you once again for your valuable feedback.

---

### Meta-Review · Area_Chair_CCAi · 2024-12-20

**Metareview:**

This paper provides a new benchmark for fixed wing aircrafts solved as multi-goal long horizon RL, with velocity control. The authors provide an easy to use benchmark, demonstrations, RL and IRL algorithms and perform a detailed empirical study.

Strengths:
Reasonable benchmark which is not saturated, and potentially useful for the GCRL and RL community
Well designed benchmark with lots of baselines and ablations

Weaknesses:
Given this is not a typically used benchmark, more motivation is needed on why this benchmark should be adopted.
A larger range of tasks could be useful

Overall this benchmark was well appreciated by the reviewers, seem like a benchmark to add to the set of standard GCRL evals. I would suggest the authors work with farama foundation and Gymnasium or with other folks to incorporate this into standard evals.

**Additional Comments On Reviewer Discussion:**

The reviewers did have questions about why this benchmark is general and broadly applicable, how it compares to other fixed wing benchmark, what the demos look like and also how this compares to standard methods. Moreover they made some great suggestions about how to make the system more usable. Reviewer ecAp brought up concerns about writing quality, but did not make concrete and actionable suggestions. This was downweighted.

---

### Decision · Program_Chairs · 2025-01-22

Accept (Poster)